# Two years with extreme and little snowfall: Effects on energy partitioning and surface energy exchange in a high-Arctic tundra ecosystem

C. Stiegler[1], M. Lund[1,2], T.R. Christensen[1,2], M. Mastepanov[1,2], A. Lindroth[1]

[1]Department of Physical Geography and Ecosystem Science, Lund University, Lund, Sölvegatan 12, 223 62 Lund, Sweden
[2]Arctic Research Centre, Department of Bioscience, Aarhus University, Roskilde, Frederiksborgvej 399, 4000 Roskilde, Denmark

*Correspondence to*: C. Stiegler (christian.stiegler@nateko.lu.se)

**Abstract.** Snow cover is one of the key factors controlling Arctic ecosystem functioning and productivity. In this study we assess the impact of strong variability in snow accumulation during two subsequent years (2013-2014) on the land-atmosphere interactions and surface energy exchange in two high-Arctic tundra ecosystems (wet fen and dry heath) in Zackenberg, Northeast Greenland. We observed that record-low snow cover during the winter 2012/13 resulted in strong response of the heath ecosystem towards low evaporative capacity and substantial surface heat loss by sensible heat fluxes ($H$) during the subsequent snow melt period and growing season. Above-average snow accumulation during the winter 2013/14 promoted summertime ground heat fluxes ($G$) and latent heat fluxes ($LE$) at the cost of $H$. At the fen ecosystem a more muted response of $LE$, $H$ and $G$ was observed in response to the variability in snow accumulation. Overall, the differences in flux partitioning and in the length of the snow melt periods and growing seasons during the two years had a strong impact on the total accumulation of the surface energy balance components. We suggest that in a changing climate with higher temperature and more precipitation the surface energy balance of this high-Arctic tundra ecosystem may experience a further increase in the variability of energy accumulation, partitioning and redistribution.

## 1 Introduction

The presence or absence of snow has a strong impact on the land-atmosphere interactions and on the exchange of energy and mass. The high albedo of snow (Warren, 1982) reduces the amount of absorbed shortwave radiation at the surface which generally leads to a smaller magnitude of the surface energy balance components. The influence of the snow on the energy balance is most pronounced during spring when the commonly patchy distribution of snow causes strong spatial variations in surface temperature and surface energy balance components (Chernov, 1988). The meltwater in Arctic soils contributes a considerable proportion of plant available water during summertime and as such, end-of-winter snow depth constitutes an important control of the summertime energy partitioning into sensible and latent heat fluxes (Langer et al., 2011).

Since the end of the Little Ice Age the climate in the Arctic has undergone a substantial warming to the highest temperatures in 400 years (Overpeck et al., 1997). Warming has further accelerated during the second half of the 20th century and was almost twice as strong as the global average (Stocker et al., 2013). Between the years 1966 and 2003 temperatures in the Arctic increased by 0.4°C per decade with most pronounced warming during the cold seasons (McBean et al., 2005). A reanalysis of meteorological observations over the period 1989-2008 shows that near-surface warming is 1.6°C during autumn and winter and 0.9°C and 0.5°C during spring and summer (Screen and Simmonds, 2010). It is suggested that diminishing sea ice, snow- and ice-albedo feedbacks and atmospheric energy transport into the Arctic govern Arctic temperature amplification (Graversen et al., 2008; Screen and Simmonds, 2010; Bintanja and van der Linden, 2013)

Precipitation in the Arctic is generally low, however, for the period from 1900 to 2003 precipitation increased by 1.4% per decade (McBean et al., 2005) with a pronounced increase mostly during winter (Becker et al., 2013). The observed contribution of snow precipitation to total annual precipitation has declined (Hartman et al., 2013). Extreme events such as extremely high temperatures and heavy precipitation have increased while extremely low temperatures have decreased over most parts of the Arctic (Hartman et al., 2013). During the 21st century the ongoing changes in the temperature and precipitation regime are expected to continue. By the end of the century, models based on the Representative Concentration Pathways (RCP) 4.5 scenario predict an average warming of 3.9°C over Arctic land areas (Stocker et al., 2013) and an increase in precipitation of more than 50%, mostly during autumn and winter (Bintanja and Selten, 2014). However, due to the increase in air temperature and rain-on-snow events the maximum amount of snow accumulation on the ground is projected to increase by only 0-30% and snow cover duration might decrease by 10-20% over most of the Arctic regions (Callaghan et al., 2011a).

At our study region in Zackenberg, Northeast Greenland, mean July air temperatures during the period 2000-2010 increased by 0.18°C yr$^{-1}$ and the active layer thickened by 1.5 cm yr$^{-1}$ (Lund et al., 2014). Based on the IPCC SRES A1B scenario (Nakićenović et al., 2000), local climate modelling for the region predict an increase in mean annual air temperature by 4.1°C for the period 2051-2080 compared to 1961-1990 with highest increase during winter (6.6°C) and spring (7.4°C) while precipitation over eastern Greenland is projected to increase by 60% (Stendel et al., 2007).

Arctic ecosystems are highly adapted to extreme seasonal variability in solar radiation, temperature, snow cover and precipitation. However, studies have shown that winter warming events and interannual snow cover variability affect ecosystem functioning in various adverse ways and these extremes are expected to occur more frequently in the future (Callaghan et al., 2005; Kattsov et al., 2005; Stocker et al., 2013). Hence, there is an urgent need to assess their impact on Arctic ecosystems. Several studies have focused on the effect of extreme temperatures on plant productivity and carbon sequestration (Chapin III et al., 1995; Marchand et al., 2005; Euskirchen et al., 2006; Bokhorst et al., 2008; Bokhorst et al., 2011) but the direct impact of successive snow cover variability on the land-atmosphere interactions and surface energy balance components is largely unknown.

Here we examine the impact of strong variability in snow accumulation by studying the land-atmosphere interactions and surface energy balance components in a high-Arctic tundra heath and fen environment during two subsequent years (2013-2014) with distinct differences in end-of-winter snow depth. Our study area is located in Zackenberg in Northeast Greenland

where record-low snow accumulation was observed during the winter 2012/13 (Mylius et al., 2014), followed by snow-rich conditions during the winter 2013/14. This sequence of strong variability in snow accumulation forms the following objectives of our study: (1) To assess the magnitude of the energy balance components and moisture exchange during the snow melt periods and growing seasons in 2013 and 2014, and (2) to quantify and evaluate the driving factors of surface energy partitioning during the observation period at our high-Arctic fen and heath site.

## 2 Materials and methods

### 2.1 Study sites

The study sites are located in the valley Zackenbergdalen in Northeast Greenland near the Zackenberg Research Station (74°30'N, 20°30'W) (Fig. 1a). The valley is surrounded by mountains to the west, north and east while the Young Sound and Tyrolerfjord form the valley boundary to the south. Vegetation is sparse and mainly found in the valley bottom and on the lower parts of the slopes. *Cassiope* heaths, *Salix arctica* snow-beds, grasslands and fens with sedges and grasses dominate in the lowlands while open *Dryas* sp. heaths and grasslands form the main plant communities on the slopes (Bay, 1998).

We conducted measurements of surface energy balance components and meteorological variables in a wet fen and in a tundra heath, with a distance of approx. 600 meters between the two measurement towers. The fen area can be divided into a continuous fen, with flat areas dominated by *Eriophorum scheuchzeri*, *Carex stans* and *Duponita psilosantha*, and a hummocky fen dominated by *E. triste*, *S. arctica* and *Andromeda latifolia* (Bay, 1998; Elberling et al., 2008). The tundra heath site is characterized by *Cassiope tetragona*, *D. integrifolia*, *Vaccinium uliginosum* and patches of mosses, *E. scheuchzeri* and *S. arctica* (Lund et al., 2012).

Since August 1995 meteorological and environmental monitoring activities have been conducted by the Zackenberg Ecological Research Operations (ZERO), a part of the Greenland Ecosystem Monitoring (GEM) programme. Mean annual air temperature in Zackenberg (1996-2013) is -9.0°C, with an average span from -19.3°C in January to +6.3°C in July (Mylius et al., 2014). Annual precipitation is low (211 mm) (Mylius et al., 2014) and approx. 85% consists of snow precipitation (Hansen et al., 2008). Snow cover is unevenly distributed in the valley with large deposits on south-facing slopes as winds from the north (offshore) dominate during the winter (Soegaard et al., 2001). During the growing season winds from south-east (onshore) are dominating (Elberling et al., 2008). The area is located within the zone of continuous permafrost and active layer thicknesses at the end of the summer reach between 0.4 and 0.8 m within the valley (Christiansen, 2004; Pedersen et al., 2012).

### 2.2 Measurements

Fluxes of sensible ($H$) and latent heat ($LE$) at the wet fen and the dry heath were measured by two eddy covariance systems. Standard flux community instrumentation and processing schemes were used to ensure reliable data quality. The eddy covariance system at the wet fen, e.g., uses well-accepted measurement standards of ICOS (Integrated Carbon Observation System). At the wet fen a 3 m tower was equipped with a LI-7200 (LI-COR Inc., USA) enclosed-path gas analyser and a Gill

HS (Gill Instruments Ltd, UK) 3D wind anemometer. Air was drawn at a rate of 15 L min$^{-1}$ through a 1 m long tube (9 mm inner diameter). Data from both sensors was sampled at a rate of 20 Hz and fluxes were calculated using EddyPro software (LI-COR Inc., USA). Air temperature ($T_a$), relative humidity ($RH$) and air pressure measured by external sensors was used in flux calculations. The gas analyser was calibrated based on manual calibrations using air with known $CO_2$ concentration and based on estimated $H_2O$ concentration from $T_a/RH$ measurements. Post-processing and quality checks follow standard procedures (Aubinet et al., 2012).

At the dry heath a 3 m tower was equipped with a Gill R3 (Gill Instruments Ltd, UK) sonic anemometer and a LI-7000 (LI-COR Inc., USA) gas analyser. Air was drawn through 6.2 m of tubing (inner diameter: 1/8'') at a rate of 5.5 L min$^{-1}$ to the sensor. To ensure that the eddy covariance measurements capture all scales of mixed-layer turbulence, cospectral analysis (Wyngaard and Cote, 1972) between the vertical wind velocity and turbulent energy flux was performed at both study locations. Data processing for both study locations is further summarized in Soegaard et al., 2001 and Lund et al. (2012, 2014). At both locations snow depth measurements (SR50A, Campbell Scientific, USA) were used to dynamically estimate sonic height above the snow layer and soil temperature (T107, Campbell Scientific, USA) at a depth of 2, 10, 20, 40, 50 and 60 cm, soil heat flux (HFP01 Hukseflux, The Netherlands) at a depth of 4 cm, net radiation (CNR4 Kipp & Zonen, The Netherlands) at a height of 3 m and snow pack temperature at 10, 20, 40, 60, 90 and 120 cm above the soil surface were measured. At the dry heath, soil moisture (SM 300, Delta-T Devices, UK) was measured at a depth of 5, 10, 30 and 50 cm.

Ancillary meteorological parameters such as air temperature and humidity (HMP 45D, Vaisala, Finland), radiation components (CNR1, Kipp & Zonen, The Netherlands) and precipitation (52203, R. Young Company, USA) are provided from a nearby meteorological station operated by Asiaq – Greenland Survey. This station is located at the same heath, approx. 150 m away from the heath eddy covariance tower. Normalized Difference Vegetation Index (NDVI) was measured at the wet fen site (SKR 1800, Skye Instruments Ltd, UK). Power supply for all stations was provided by diesel generators from the nearby Zackenberg Research Station (May to October), and solar panels and a wind mill (Superwind 350, superwind GmbH, Germany) during the period when the research station was closed.

### 2.3 Data analysis and derived parameters

The surface energy balance of the wet fen and the dry heath is described by:

$$R_{net} = H + LE + G + E_{melt} \tag{1}$$

where $R_{net}$ is the net radiation, $H$ is the sensible heat flux, $LE$ is the latent heat flux, $G$ is the ground heat flux at the soil or snow surface, and $E_{melt}$ is the energy flux used for snow melt (Table 1). During the snow-free season $E_{melt}$ is zero. When directed away from the surface, $H$, $LE$ and $G$ are positive and $R_{net}$ is negative. Bowen ratio ($H/LE$) and ratios of $H/R_{net}$, $LE/R_{net}$ and $G/R_{net}$ are used to characterize relative magnitude of the heat transfer from the surface.

The net radiation balance ($R_{net}$) was defined as:

$$R_{net} = RS\downarrow + RL\downarrow - (RS\uparrow + RL\uparrow) \tag{2}$$

where $RS\downarrow$ and $RS\uparrow$ are incoming and outgoing shortwave radiation, and $RL\downarrow$ and $RL\uparrow$ are upwelling and downwelling longwave radiation, respectively. When directed away from the surface all radiative components are negative. Surface albedo was calculated as the quotient between $RS\downarrow$ and $RS\uparrow$. Missing values from radiative components at the dry heath site were filled with measurements from the nearby Asiaq meteorological tower.

Ground heat flux at the soil or snow surface ($G$) was calculated by adding the storage flux in the layer above the heat flux plate ($S$) to the measured flux:

$$S = C_s \frac{\Delta T_s}{\Delta t} d \tag{3}$$

where $\Delta T_s/\Delta t$ is the change in soil or snow pack temperature (K) over time $t$ (s), $d$ is the heat flux plate installation depth (m) and $C_s$ is the soil or snow pack heat capacity (J m$^{-3}$ K$^{-1}$) defined as:

$$C_s = \rho_b C_d + \theta_v \rho_w C_w \tag{4}$$

where $\rho_b$ is the bulk density, $C_d$ is the dry soil heat capacity of 840 J kg$^{-1}$ K$^{-1}$ (Hanks and Ashcroft, 1980) or the heat capacity of ice (2102 J kg$^{-1}$ K$^{-1}$), $\theta_v$ is the volumetric soil or snow pack water content (m$^3$ m$^{-3}$), $\rho_w$ is the water density (1000 kg m$^{-3}$) and $C_w$ is the water heat capacity (4186 J kg$^{-1}$ K$^{-1}$). For $\rho_b$ a value of 900 kg m$^{-3}$ at the heath and 600 kg m$^{-3}$ at the fen (Elberling et al., 2008) was used during the growing season while during the snow melt period in 2014 the density of the snow pack was

derived from in-situ measurements. Since no snow density measurements were performed during the snow melt period in 2013 and the soil surface was not completely snow-covered during that period, $G$ in 2013 was not corrected for heat storage within the snow pack.

The aerodynamic resistance ($r_a$, s m$^{-1}$) determines the turbulent heat transfer from the surface and was defined as (Monteith and Unsworth, 2013):

$$r_a = \frac{u}{u_*^2} + 6.2 u_*^{-0.67} \tag{5}$$

where $u$ is the wind speed (m s$^{-1}$) and $u_*$ is the friction velocity (m s$^{-1}$).

Surface resistance ($r_s$, s m$^{-1}$), as a measure to quantify the stomatal control in the canopy on the turbulent fluxes, was calculated as (Shuttleworth, 2007):

$$r_s = \left(\frac{\Delta}{\gamma}\beta - 1\right) r_a + (1 + \beta)\frac{\rho c_p}{\gamma}\frac{D}{A} \tag{6}$$

where $\Delta$ is the slope of the saturated vapour pressure curve (Pa K$^{-1}$), $\gamma$ is the psychrometric constant (Pa K$^{-1}$), $\beta$ is the Bowen ratio, $\rho$ is the air density (kg m-$3$) $c_p$ is the specific heat capacity of air at constant pressure (J kg$^{-1}$ K$^{-1}$), $D$ is the atmospheric vapour pressure deficit (Pa) and $A$ is the available energy for evaporation ($R_{net} - G$, W m$^{-2}$).

The decoupling coefficient ($\Omega$) (Jarvis and McNaughton, 1986) expresses the degree of interaction between $r_a$ and $r_s$:

$$\Omega = \left(1 + \frac{\Delta}{\Delta+\gamma}\frac{r_s}{r_a}\right)^{-1} \tag{7}$$

The decoupling coefficient varies from 0 to 1 where $\Omega$ close to zero indicates a strong coupling between the vegetation and the atmosphere, with vapour pressure deficit ($VPD$) being the main driver of $LE$, whereas $\Omega$ close to 1 suggest a decoupling of $LE$ and $VPD$, with $R_{net}$ being the main driver for $LE$.

Priestley-Taylor coefficient ($\alpha$) (Priestley and Taylor, 1972) was calculated as:

$$\alpha = \frac{\Delta + \gamma}{\Delta(1+\beta)} \tag{8}$$

Over ocean and saturated land surfaces the dimensionless α equals 1.26 but fluctuates depending on surface structure and meteorological conditions.

To assess the full impact of differences in snow accumulation on the surface energy exchange the presented results focus on four major subperiods, i.e. polar night, pre-melt season, snow melt period and growing season, which we defined the following: Polar night is the time period when the sun is below the free horizon. In Zackenberg, the polar night lasts from 10 November to 4 February (86 days). The subsequent pre-melt season marks the time period between the polar night and the first day of snow melt. In spring, a steady and constant decrease in daily average albedo and snow cover thickness until snow cover

diminished was defined as the snow melt period. The beginning of the subsequent growing season was defined as the time when albedo was lower than 0.2. Positive daily average air temperature and top soil layer temperature (2 cm depth), net radiation >0 W m$^{-2}$ and albedo lower than 0.2 define the growing season.

Gap-filling of the eddy covariance data was performed based on a look-up table approach (Falge et al., 2002; Reichstein et al., 2005) using the REddyProcWeb online tool (Max Planck Institute for Biogeochemistry). Due to very limited data availability

at both locations during the second snow melt period in 2014 no gap-filling of $H$ and $LE$ was applied and turbulent fluxes of $H$ and $LE$ were excluded from the analysis during that period.

## 3 Results

### 3.1 Energy balance closure

The observed slopes of the regressions between available energy at the surface ($R_{net} - G$) and the sum of the turbulent heat

fluxes ($H+LE$) serve as an indicator for the energy balance closure. Figure 2 shows the comparison between mean daily $R_{net} - G$ and mean daily $H+LE$ for the two study sites. The observed slopes at the wet fen are 0.68 during both 2013 and 2014. For the dry heath the slopes are 0.81 (2013) and 0.82 (2014). Possible reasons for the observed energy imbalance at both sites are discussed in section 4.1.

### 3.2 Polar night and pre-melt season

The pattern of snow accumulation during the polar night and pre-melt season differed strongly between the two winters (Fig. 1b). There was no distinct development of a closed snow cover or major events of snow accumulation during the first winter (2012/13) and by the end of the pre-melt season as little as 0.09 m of snow pack at the wet fen and 0.14 m at the dry heath was present. The second winter (2013/14) showed a similar trend in the beginning of the winter with snow pack thickness of less than 0.1 m until mid-December 2013. After that, a snow pack of 0.8 m developed during two major events of snow fall (17-

25 December 2013 and 21-27 January 2014) and by the end of the second winter the snow pack reached a thickness of 1.04 m at the wet fen and 0.98 m at the dry heath (Fig. 3).

In the absence of solar radiation during the polar night thermal radiation ($RL_{net}$) is the sole driver of the surface energy balance (Fig. 3). Daily average downwelling longwave radiation ($RL\downarrow$) was 193 W m$^{-2}$ during the first polar winter (2012/13) and 213 W m$^{-2}$ during the second polar winter (2013/14). The corresponding values for the upwelling longwave radiation ($RL\uparrow$) were 219 W m$^{-2}$ and 237 W m$^{-2}$ for the two polar winter periods. The differences in longwave radiation corresponded well with the differences in air and snow surface temperatures. The average air temperature was -19.2°C and -15.2°C for the two consecutive polar winter periods and the snow surface temperature was -24.1°C and -19.2°C, respectively. Overall, average $RL_{net}$ was -27 W m$^{-2}$ in 2012/13 and -24 W m$^{-2}$ in 2013/14. The lowest air temperatures (-33.7°C and -33.7°C) and snow surface temperatures (-39.8°C and -39.9°C) for both polar winters, however, were reached shortly after the onset of the pre-melt seasons. In 2013, this coldest period was followed by a foehn event with air temperatures just below -1°C on 7 March. Incoming solar radiation ($RS\downarrow$) showed a continuous increase over the pre-melt season, with higher values observed during the winter 2012/13 compared to 2013/14, mainly driven by differences in surface albedo.

### 3.3 Snow melt season

At the beginning of the snow melt period in 2013 the vegetation on both sites was not completely snow-covered. Snow melt started on 13 May and lasted until 29 May (17 days) at the wet fen and until 30 May (18 days) at the dry heath. Snow ablation in 2014, however, started 7 days earlier compared to 2013 (6 May) but lasted 28 days longer at the dry heath site (27 June) and 25 days longer at the wet fen site (23 June). Daily average air temperatures remained below 0°C for most of the snow melt periods with average values of -1.8°C in 2013 and -1.3°C in 2014 (Fig. 4). High rates of snow ablation after 6 June 2014 coincided with daily average air temperatures >0°C which peaked on 16 June 2014 with 9.6°C.

During the snow melt period in 2013, albedo decreased gradually from about 0.76 in the beginning of the period down to 0.15 for the wet fen and 0.10 for the dry heath (Fig. 4). The decrease in albedo happens simultaneously with a continuous increase in net radiation ($R_{net}$) (Fig. 4). However, while daily average $R_{net}$ remained negative for most of the first half of the snow melt period in 2014 (Fig. 4), positive values dominated the entire snow melt period in 2013. Substantial differences in albedo cause these variations, especially in the beginning of the snow melt seasons when daily average albedo was >0.60. During that period, average albedo at both sites was 0.71 in 2013 and 0.79 in 2014. These differences in the surface characteristics reduced average $R_{net}$ in 2014 by 4.1 W m$^{-2}$ although incoming solar radiation ($RS\downarrow$) was 87.0 W m$^{-2}$ higher compared to 2013. By the end of the snow melt period in 2013, $R_{net}$ accumulated 76.1 MJ m$^{-2}$ at the dry heath and 94.3 MJ m$^{-2}$ at the wet fen. The corresponding values for 2014 were 174.5 MJ m$^{-2}$ and 202.9 MJ m$^{-2}$, respectively. Due to the lower albedo during the first snow melt season the soil experienced a faster spring warming compared to 2014. Top-soil layer temperatures ($T_{surf}$) at the dry heath increased by 0.41°C d$^{-1}$ in 2013 and 0.28°C d$^{-1}$ in 2014. At the wet fen, the warming of the soil was 0.35°C d$^{-1}$ in 2013 and 0.28°C d$^{-1}$ in 2014.

Daily average sensible heat flux ($H$) was small and negative (-4.1 W m$^{-2}$) during the first ten days of the snow melt period in 2013 at the wet fen while it showed slightly positive values (4.3 W m$^{-2}$) at the dry heath (Fig. 4). The latent heat flux ($LE$) increased gradually up to about 18 W m$^{-2}$ at the wet fen and up to 23 W m$^{-2}$ at the dry heath (Fig. 4). During this initial period the net radiation increased gradually up to about 58 W m$^{-2}$ at the wet fen and 43 W m$^{-2}$ at the dry heath. Over the entire snow melt period, $H$ dominated over $LE$ at the dry heath (Bowen ratio 1.7) with mean daily average $H$ and $LE$ of 25.0 W m$^{-2}$ and 14.9 W m$^{-2}$, respectively, while at the wet fen $LE$ dominated over $H$ (Bowen ratio 0.5), with average $H$ and $LE$ of 7.6 W m$^{-2}$ and 15.6 W m$^{-2}$ (Table 3). By the end of the snow melt period $H$ accumulated 32.5 MJ m$^{-2}$ at the dry heath and 3.5 MJ m$^{-2}$ at the wet fen. The corresponding values for $LE$ were 22.7 MJ m$^{-2}$ and 21.1 MJ m$^{-2}$, respectively. This corresponds to accumulated evaporation of 9.3 mm and 8.6 mm of evaporated water during the entire spring snow melt at the dry heath and the wet fen.

During both years the ground heat flux ($G$) at the wet fen and the dry heath used ~10-15% of the total energy supplied by $R_{net}$. The prolonged snow melt period during the second year had no impact on the magnitude of $G$. However, the total amount of energy supplied to the ground was larger in 2014 than in 2013. At the dry heath, average $G$ of 5.1 W m$^{-2}$ in 2013 and 5.1 W m$^{-2}$ in 2014 resulted in a total energy consumption of 8.0 MJ m$^{-2}$ in the first year and 20.5 MJ m$^{-2}$ in the second year. Similar behaviour was observed at the wet fen where average $G$ of 8.4 W m$^{-2}$ in 2013 and 7.2 W m$^{-2}$ in 2014 added up to a total energy consumption of 10.2 MJ m$^{-2}$ and 28.6 MJ m$^{-2}$. The average latent heat content of snow cover during the snow melt period in 2014 was 3.1 MJ m$^{-2}$ which equals to 1.8% of $R_{net}$ at the dry heath and 1.5% of $R_{net}$ at the wet fen.

### 3.4 Growing season

### 3.4.1 Season length and meteorological conditions

The growing season in 2013 lasted 101 days at the wet fen (30 May – 8 September) and 100 days at the dry heath (31 May – 8 September). Compared to 2013, the season in 2014 was 30 days shorter on the wet fen (24 June – 4 September) and 31 days shorter at the dry heath (28 June – 4 September). The length of the growing season in the first year clearly exceeds the average 2000-2010 length (78 days) while the second year shows below-average length of the growing season (Lund et al., 2014). Mean July-August air temperatures ($T_a$) were similar in both years, reaching 6.5°C in the first year and 6.3°C in the second year, mean $T_a$ over the entire growing seasons were lower in 2013 (5.3°C) compared to 2014 (6.1°C) (Fig. 5). Growing season air temperature in 2013 was slightly below the 2000-2010 average (5.5°C) (Lund et al., 2014) and above-average in 2014. Total amount of precipitation during both years clearly exceed the 2000-2010 average of 27.2 mm (Lund et al., 2014), reaching 80 mm in 2013 and 65.5 mm in 2014. In 2013, pronounced events of rainfall (>5 mm d$^{-1}$) occurred at the end of the growing season while in the second year precipitation mainly fell shortly after snow melt and in late-August (Fig. 5).

### 3.4.2 Radiation balance

During the growing season in 2013, mean daily $R_{net}$ was slightly higher at the wet fen (114.2 W m$^{-2}$) compared to the dry heath (111.4 W m$^{-2}$) (Table 4) while during July-August both sites showed similar $R_{net}$ (~93 W m$^{-2}$). Compared to 2013, mean daily

$R_{net}$ was slightly higher during July-August 2014, with 94.3 W m$^{-2}$ at the wet fen and 98.6 W m$^{-2}$ at the dry heath, but lower over the entire growing season, with 100.2 W m$^{-2}$ and 98.9 W m$^{-2}$, respectively (Table 4, Fig. 6). By the end of the season in 2013, accumulated $R_{net}$ was 1006.6 MJ m$^{-2}$ at the wet fen and 967.4 MJ m$^{-2}$ at the dry heath compared to 632.2 MJ m$^{-2}$ and 589.5 MJ m$^{-2}$ in 2014. This increase of accumulated $R_{net}$ by 59% (fen) and 64% (heath) in 2013 relates to the earlier onset of the growing season. By 24 June 2013, accumulated $R_{net}$ at the wet fen reached 406 MJ m$^{-2}$ during the first 25 days of the growing season while in 2014, the growing season started on 24 June and the surface was thus still snow covered until that date. Similar values were observed at the dry heath (423 MJ m$^{-2}$).

In 2013, mean surface albedo at both sites increased gradually during the growing season and reached its maximum towards the end of the season. Over the entire growing season albedo was higher at the wet fen (0.20) compared to the dry heath (0.16). During a snow fall event on 30 August 2013 albedo increased up to 0.32. Similar trends in albedo were observed in 2014 but mean surface albedo over the entire growing season was lower compared to 2013 with 0.17 at the wet fen and 0.13 at the dry heath. The differences between the two years are more pronounced during the period July-August when mean albedo was 0.20 (2013) and 0.16 (2014) at the wet fen and 0.16 (2013) and 0.12 (2014) at the dry heath. The pronounced visible differences in greenness during the two growing seasons (Fig. 1b) was also reflected in average NDVI, with higher values observed in 2014 (0.49) compared to 2013 (0.45) (Table 4).

### 3.4.3 Turbulent heat fluxes

The magnitude of the turbulent heat fluxes and the total amount of accumulated energy by $H$ and $LE$ revealed remarkable differences between the two years (Fig. 6, Fig. 7). Over the entire growing season in 2013, average fluxes of $H$ and $LE$ used 56% (62.6 W m$^{-2}$) and 14% (16.0 W m$^{-2}$), respectively, of $R_{net}$ at the dry heath. The corresponding values for the wet fen were 37% (41.5 W m$^{-2}$) and 30% (25.2 W m$^{-2}$), respectively (Table 4). This clear dominance of $H$ over $LE$ at both locations is also reflected in the average Bowen ratio ($H/LE$) which reached a maximum of 3.9 at the dry heath and 1.6 at the wet fen.

During 2014, $LE$ at the dry heath used 26% (29.3 W m$^{-2}$) of $R_{net}$ (Table 4). This corresponds to an almost doubling of $LE$ compared to 2013. Mean fluxes of $H$ at the dry heath were 36.6 W m$^{-2}$ and $H$ used 37% of $R_{net}$. Average Bowen ratio reached 1.3 which indicates a growing importance of $LE$ at the dry heath compared to 2013. At the wet fen, mean daily $LE$ was 23.3 W m$^{-2}$ and $LE$ used 25% of $R_{net}$. The corresponding value for $H$ was 26.6 W m$^{-2}$ and $H$ used 29% of $R_{net}$. Average Bowen ratio reached 1.1.

From the first day after snow melt until the last day of the growing season in 2013 accumulated $H$ was 546.2 MJ m$^{-2}$ at the dry heath and 365.7 MJ m$^{-2}$ at the wet fen while accumulated $LE$ was 139.9 MJ m$^{-2}$ and 222.0 MJ m$^{-2}$, respectively (Fig. 7). By the end of the growing season in 2014, however, accumulated $H$ was 220.2 MJ m$^{-2}$ at the dry heath and 167.8 MJ m$^{-2}$ at the wet fen. The corresponding values for $LE$ were 174.7 MJ m$^{-2}$ and 146.9 MJ m$^{-2}$, respectively (Fig. 7). The observed values of accumulated $H$ at the dry heath during 2014 correspond to a decrease in accumulated $H$ by 60% and an increase in accumulated $LE$ by 24% compared to 2013. At the wet fen, accumulated $H$ and $LE$ in 2014 correspond to a decrease by 44% and 33%, respectively, compared to 2013.

The total accumulated evapotranspiration (*ET*) differed significantly between the seasons (Fig. 8). At the wet fen, total *ET* reached 91 mm during the growing season in 2013 and 48 mm in 2014. These values correspond to ~114% and ~73% of the total precipitation during the growing seasons within the corresponding years. However, total *ET* at the dry heath remained below the growing season precipitation during the first growing season (57 mm, ~71%) and exceeded precipitation in the

second growing season (80 mm, ~122%).

### 3.4.4 Controls of evapotranspiration

During both growing seasons, lower wind velocity at the dry heath compared to the wet fen resulted in slightly larger aerodynamic resistances ($r_a$) at the latter site. Daily average $r_a$ showed no clear difference between the two years ranging between 75-82 s m$^{-1}$ in both years at the wet fen. The corresponding values at the dry heath were between 107-120 s m$^{-1}$ with

slightly larger variability compared to the wet fen (Fig. 9). No significant changes during the course of both growing seasons indicate that $r_a$ appeared to be independent of $RS\downarrow$ and $R_{net}$ and therefore mainly driven by atmospheric conditions.

The surface resistance ($r_s$) during both growing seasons, however, was characterized by large differences between the wet fen and the dry heath and high daily fluctuations at the latter site (Fig. 9). Daily average $r_s$ at the dry heath ranged from 93 to 923 s m$^{-1}$ in 2013 and from 64 to 428 s m$^{-1}$ in 2014, with higher mean $r_s$ in 2013 (556 s m$^{-1}$) compared to 2014 (247 s m$^{-1}$). During

the first growing season $r_s$ showed a general increase with increasing air temperature ($T_a$), ranging from ~455 s m$^{-1}$ when $T_a$ <0°C to ~804 s m$^{-1}$ when $T_a$ >12°C. During the second growing season $r_s$ decreased from ~297 s m$^{-1}$ when $T_a$ <0°C to ~210 s m$^{-1}$ when $T_a$ was between 3-6°C, followed by a slight increase to 250 s m$^{-1}$ when $T_a$ >12°C. However, no such trend was observed at the wet fen and the surface resistance had no pronounced daily fluctuations with daily averages ranging between 53 and 455 s m$^{-1}$.

The McNaughton and Jarvis decoupling factor ($\Omega$) expresses the degree of aerodynamic and radiative coupling between the vegetation and the atmosphere. The wet fen and the dry heath were characterized by mean daily $\Omega$ below or close to 0.5 during both growing seasons which indicates a relatively high contribution of vapour pressure deficit (*VPD*) on the control of *ET* (Table 4, Fig. 9). However, mean daily $\Omega$ during the first growing season was lower at the dry heath (0.30) compared to the wet fen (0.38) but showed a reverse trend during the second year with 0.52 compared to 0.48. The general decrease in $\Omega$ over

the course of the first season at the dry heath terminated around mid-August and $\Omega$ started to increase for the rest of the season. Similar behaviour for $\Omega$ was observed at the wet fen where $\Omega$ started to increase around late-August. The second growing season showed a general decrease in $\Omega$ at both sites. During both growing seasons $\Omega$ gradually decreased as $r_s$ increased. The Priestley-Taylor coefficient ($\alpha$) showed large day-to-day variations, particularly at the wet fen, ranging between 0.05 and 1.06 in 2013 and 0.21 and 1.13 in 2014. The seasonal mean of $\alpha$ was 0.60 and 0.69 with a standard error of 0.02 and 0.03 for the

respective seasons. The day-to-day variation was less pronounced at the dry heath with seasonal means of 0.44 and 0.74 and standard error of 0.02 for both years. (Table 4, Fig. 9).

### 3.4.5 Ground heat flux and soil properties

The ground heat flux ($G$) was the smallest flux in both ecosystems but due to the strong heat sink of permafrost $G$ is a vital component of the surface energy budget in this high-latitude environment. The differences in snow cover between the two years caused distinct differences in the soil water content during the two growing seasons, with strong impact on the partitioning of $R_{net}$ into $G$ at the dry heath. Average soil moisture content at the dry heath of 19% in 2013 and 34% in 2014 restrained thermal conductivity more during 2013 than during 2014. Consequently, only 5% of $R_{net}$ (5.6 W m$^{-2}$) was partitioned into $G$ in 2013 while during 2014 $G$ used 10% of $R_{net}$ (10.1 W m$^{-2}$). At the wet fen, mean daily $G$ used 12% of $R_{net}$ (10.8 W m$^{-2}$) in 2013 while in 2014, $G$ used 17% of $R_{net}$ (15.2 W m$^{-2}$).

During the course of the two growing seasons $G$ reached its maximum shortly after snow melt due to the strong thermal gradient between the soil surface and the underlying permafrost (Fig. 6). However, the low soil moisture content at the dry heath during the first growing season weakened the $G$ signal after the snow melt. No clear seasonal trend in the development of $G$ to $R_{net}$ was observed for both growing seasons and locations. By the end of the first growing season accumulated $G$ was 48.7 MJ m$^{-2}$ at the dry heath and 94.7 MJ m$^{-2}$ at the wet fen (Fig. 7). Increased thermal conductivity due to higher soil moisture content in 2014 compared to 2013 amplified the total amount of accumulated $G$ at the dry heath in the latter year although prolonged snow cover reduced the length of the second growing season. By the end of the second growing season accumulated $G$ was 60.1 MJ m$^{-2}$ at the dry heath (+23% compared to 2013) while at the wet fen, no clear difference in accumulated $G$ was observed (95.6 MJ m$^{-2}$).

### 3.5 Total surface energy balance of the snow melt season and growing season

From the first day of the beginning of the snow melt period until the last day of the growing season in 2013 $R_{net}$ accumulated 1043.5 MJ m$^{-2}$ at the dry heath and 1100.9 MJ m$^{-2}$ at the wet fen. The corresponding values for 2014 were 764.0 MJ m$^{-2}$ and 835.1 MJ m$^{-2}$, respectively. Accumulated ground heat flux for the same time period was 56.7 MJ m$^{-2}$ at the dry heath and 103.8 MJ m$^{-2}$ at the wet fen in 2013 and 83.2 MJ m$^{-2}$ and 125.3 MJ m$^{-2}$ in 2014, respectively. These values correspond to a total amount of accumulated available energy at the surface ($R_{net} - G$) of 986.8 MJ m$^{-2}$ at the dry heath and 997.1 MJ m$^{-2}$ at the wet fen in 2013. In 2014, $R_{net} - G$ was 683.4 MJ m$^{-2}$ at the dry heath and 711.8 MJ m$^{-2}$ at the wet fen which, compared to 2013, relates to a decrease in available energy at the surface by -31% and -29%, respectively.

In 2013, the amount of total $H$ loss was 578.7 MJ m$^{-2}$ at the dry heath and 369.2 MJ m$^{-2}$ at the wet fen while total $LE$ loss was 162.6 MJ m$^{-2}$ and 243.1 MJ m$^{-2}$, respectively. These values add up to a total energy consumption of $H$ and $LE$ ($H+LE$) of 741.3 MJ m$^{-2}$ at the dry heath and 612.3 MJ m$^{-2}$ at the wet fen.

## 4 Discussion

### 4.1 Energy balance closure and system performance

The observed values of energy balance closure lie in the range of other energy balance closure terms reported from various field studies in Arctic environments (Westerman et al., 2009; Langer et al., 2011; Liljedahl et al., 2011) and are in good agreement with earlier long-term observations from the dry heath (Lund et al., 2014). Studies on the closure problem of eddy covariance measurements highlight the landscape heterogeneity as a driving factor for the lack of closure (Stoy et al., 2013) but also stress the importance of measurement scales (Foken, 2008). Due to the striking small-scale heterogeneity of soil moisture availability and vegetation cover in Arctic landscapes, measurements of point-scale $G$ and radiative components may not represent the large source area of eddy covariance $H$ and $LE$. However, at our sites a flux footprint analysis at the dry heath revealed that fluxes of $H$ and $LE$ on average originate from the *Cassiope* heath (Lund et al., 2012) while the highest contribution of fluxes at the wet fen originates from the continuous fen area (Tagesson et al., 2012). Large variability in soil heat capacity due to spatial variation in soil moisture may account for underrepresentation of the stored energy between the heat flux plates and the soil surface within the eddy covariance flux footprint area (Leuning et al., 2012). Additional uncertainty in the assessment of $G$ arises from small-scale variation in active layer depth and soil thermal gradients.

Previous studies have shown large high-frequency flux attenuation of traditional closed-path gas analysers (Haslwanter et al., 2009), such as used in this study at the dry heath site, caused by tube walls and tube age (Su et al., 2004; Massman and Ibrom, 2008) and tube length (Novik et al., 2013). Enclosed-path systems, however, reduce flux attenuation compared to traditional closed-path systems due to shorter tube length (Burba et al., 2010; Novick et al., 2013). Sample screening of cospectra at our two study sites (data not shown) showed that frequency losses for water vapour and sensible heat flux were more pronounced at the wet fen (enclosed-path system) compared to the dry heath (closed-path system). At both locations, the attenuation is generally greater for water vapour flux than for sensible heat flux and at higher frequencies (n > 0.1), normalized cospectra for water vapour flux was lower at the wet fen compared to the dry heath. Interannual comparison of cospectra for both wet fen and dry heath showed similar behaviour of water vapour and sensible heat flux. Thus, we conclude that besides the heterogeneity of the tundra surfaces, high-frequency losses of both water vapour and sensible heat fluxes contribute to the observed surface energy imbalance. However, since surface energy imbalance is consistent over the two study years we are confident that the measured surface energy balance components are adequately represented for the purpose of this study.

### 4.2 Snow cover and surface energy budget

The disappearance of the snow cover and the onset of the growing season coincide with the time period when incoming solar radiation ($RS\downarrow$) is at its annual peak and the surface (soil or snow) receives high irradiance from $RS\downarrow$. During the snow melt period most of $R_{net}$ is used for warming and melting of the snow pack and therefore not available for atmospheric warming through sensible heat fluxes. This pattern was reflected in the average air temperature of the two snow melt periods which showed only small differences between 2013 and 2014 although the snow melt period in 2014 was much longer compared to

2013. The combination of snow layer thickness, snow physical properties such as density or grain size (Warren, 1982), the fraction of exposed dark underlying surface and type of vegetation (Sturm and Douglas, 2005) control the snow and soil surface albedo, with direct impact on $R_{net}$. Our results showed that low and partly absent snow cover in the winter 2012/13 resulted in a short snow melt period dominated by low albedo of the snow-soil surface and increased average $R_{net}$. Simultaneously, sensible

heat fluxes dominated at the dry heath while latent heat fluxes dominated at the wet fen. Relatively thick snow cover and a prolonged snow melt period in 2014 increased surface albedo and limited average $R_{net}$. Further, the total energy accumulated by $R_{net}$ during the snow melt period was higher in 2014 compared to 2013.

During the snow melt period, only a small fraction of $R_{net}$ can be used by plants for growth and development as the snow reflects most of the incoming solar radiation and sustains relatively low surface temperatures (Walker et al., 2001).

Consequently, a prolonged period of snow cover and snow melt at our study sites in 2014 delayed the onset of the growing season and limited the amount of total $R_{net}$ available for plant metabolism. In coherence with that, the magnitude of the surface energy balance components and the partitioning of $R_{net}$ into $H$, $LE$ and $G$ continued to be influenced by the presence of the snow. An earlier disappearance of the snow in 2013, however, was related to an earlier increase in the magnitude of $R_{net}$ at the surface which facilitated earlier plant development and growth, soil warming and permafrost active layer development through

$G$, evapotranspirative heat loss through $LE$, and atmospheric warming through $H$.

The disappearance of the snow and related increase in surface heating mark the transition into a convective summer-type precipitation regime (Callaghan et al., 2011b) which initiates a rapid and distinct change in both plant metabolism and surface energy balance. Ongoing and predicted climate change, however, promotes an increase in atmospheric moisture, winter snowfall over land areas (Rawlins et al., 2010), and variability in the amount of snowfall (Callaghan et al., 2005; Kattsov et

al., 2005; Stocker et al., 2013) while higher temperatures stimulate an earlier snow melt and transition to convective precipitation patterns (Groisman et al., 1994). We observed that the impact of this variability in snow cover and snow melt on the seasonal surface energy budget is strongly connected to the storage of meltwater in the soil and its evaporation and transpiration over the subsequent growing season. A higher proportion of soil moisture, combined with a high atmospheric moisture demand, generally stimulates $ET$. In our study, the impact of the availability of soil moisture from snow melt and its

loss through $ET$ on the surface energy budget was most pronounced at the heath. Here we observed that low soil moisture content and lack of summertime precipitation in 2013 amplified $H$ at the cost of $LE$ and $G$ while increased soil moisture in 2014 and a pronounced rainfall event in the middle of the growing season favoured $LE$ and $G$ at the cost of $H$. Further, the Bowen ratio of the heath during the same year showed that energy partitioning into $H$ and $LE$ was similar to the wet fen. In contrast, the wet fen showed attenuated behaviour of $LE$, $ET$, $H$ and $G$ to the variability in snow meltwater as the fen receives

its moisture supply mostly from minerotrophic water supply which remained relatively stable over the two study years. Growing season variability in the partitioning of the surface energy balance components was also observed at a polygonal tundra site in Siberia (Boike et al., 2008). However, the observed differences were mainly driven by variability in summertime precipitation.

Negative water balances during the growing season, with *ET* exceeding precipitation, are common in high-latitude ecosystems (Woo et al., 1992). Depending on the type of water supply, tundra ecosystems may therefore experience differential responses to climate variability and climate change (Rouse, 2000; Boike et al., 2008). At the wet fen, our observations showed negative water balances during the growing season 2013 but the loss of water through *ET* was compensated by moisture supply from

soil moisture. Consequently, the partitioning of $R_{net}$ into *LE* showed only small differences between the two growing seasons. At the dry heath, the growing season in 2013 ended with a positive water balance. However, this was related to both pronounced precipitation at the end of the season and to low rates of *ET* due to declining $R_{net}$. During most of the season snow meltwater was the only supplier of soil moisture and the small amount of snow meltwater was evaporated relatively soon after snow melt. Consequently, for most parts of the remaining season the soil was not able to supply moisture for *ET*. This resulted in low *LE*,

relatively low soil thermal conductivity and *G*, and in a clear dominance of *H*. During the growing season in 2014, the water balance of the heath was negative over the entire season but the soil experienced greater saturation from snow meltwater storage which was reflected in the partitioning of the surface energy balance towards a greater share of *LE* and *G* on $R_{net}$.

The effects of *VPD*, soil moisture and air temperature on plant stomata and the impact of $r_s$ on *ET* and *LE* has been documented for a large number of species and ecosystems (Losch and Tenhunen, 1983; Lafleur and Rouse, 1988; Kasurinen et al., 2014).

At the dry heath, the observed values of $r_s$ suggest strong vegetation response to the different regimes of snow and surface wetness in the two study years compared to previous studies (Soegaard et al. 2001; Lund et al., 2014), while at the wet fen attenuated behaviour of $r_s$ was observed. Arctic ecosystems are generally characterised by a high proportion of free water and mosses which, unlike vascular plants, limit moisture transfer during high *VPD*. However, the concept of surface resistance neglects this contribution of free water and non-vascular plants (Kasurinen et al., 2014) and therefore the application of $r_s$ and

$\Omega$ is difficult in Arctic environments. The seasonal differences in $r_s$ at the dry heath may also be explained by the characteristics of the growing season precipitation regimes and water content of the moss-soil layer (McFadden et al., 2003). Bowen ratio, $r_s$ and $\Omega$ were closely-coupled to precipitation, resulting in a decrease in Bowen ratio, $r_s$ and $\Omega$ during periods of rainfall while longer periods without precipitation resulted in high values of Bowen ratio, $r_s$ and $\Omega$. This behaviour for all parameters was more pronounced in 2013 compared to 2014. In 2013, the combined effects of low soil moisture, lack of precipitation during

the period with relatively high *RS↓* and high *VPD* may be responsible for the observed magnitudes.

Variability of snow cover and length of the growing season is not only reflected in the partitioning of the surface energy balance components. More important, total accumulated $R_{net}$ increases with increased length of the snow-free period. Further, any increase in the length of the snow-free season or in summer temperatures is manifested in a general increase in *ET*, resulting in negative water balances over the snow-free season if precipitation shows no increase (Eugster et al., 2000). Our results

showed that with the onset of the snow melt period and growing season during the part of the year when incoming solar radiation is at its peak, an earlier onset of the growing season of approx. four weeks resulted in dramatic differences in accumulated energy fluxes at the end of the growing season. Summarizing, the increased amount of accumulated $R_{net}$ contributed to increased *ET* and surficial heat loss from *LE*, soil and permafrost warming through *G*, and atmospheric warming through *H*.

# 5 Conclusions and outlook

In this study we documented the effects of variability in snow accumulation on the surface energy balance of a high-Arctic tundra ecosystem in Northeast Greenland during two subsequent years (2013-2014). The most important findings include:

- Low snow cover during the winter 2012/13 promoted low surface albedo and positive daily average $R_{net}$ over the snow melt period in 2013 while extensive snow cover during the winter 2013/14 resulted in high albedo and reduced $R_{net}$ during the snow melt period in 2014.

- The heath's energy budget was strongly affected by the variability in snow cover, resulting in substantial heat loss by $H$ at the cost of $LE$ and $G$ in 2013 while in 2014, $LE$ and $G$ showed a strong increase at the cost of $H$. In contrast, the wet fen showed attenuated response to the variability in snow cover due to differences in the local hydrological settings.

- At both sites, the variation in the length of the snow melt periods and growing seasons was manifested in substantial differences in the total amount of accumulated energy balance components.

Among the research community, mean values of the climatic site conditions, surface energy and carbon exchange have been regarded as powerful indicators for ecosystem productivity and model evaluation is widely based on these parameters. However, the frequency and magnitude of weather extremes and the pace of ongoing climate change are major challenges for ecosystems all over the planet. In the Arctic and subarctic, the closely-coupled hydrological, biological and soil thermal regimes respond nonlinearly to a myriad of controlling factors (Liljedahl et al., 2011). It is therefore essential to further incorporate such data of weather extremes into research and modelling tools (Jentsch et al., 2007), especially in the Arctic and subarctic, where the lack of observational studies result in a blur picture of high-latitude ecosystem response and contribution to climate change. The results of this study may be a valuable contribution for modelling tools as it is, to our knowledge, the first study that evaluates the impact of successive snow cover variability on the land-atmosphere interactions and surface energy balance components in Greenlandic tundra ecosystems.

## Author contribution

The original idea for the paper was suggested by C.S., M.L. and A.L. and discussed and developed by all authors. M.L. and C.S. performed the data analysis. C.S. prepared the manuscript with contributions from all co-authors.

## Acknowledgements

We thank GeoBasis programme for running and maintaining the flux measurement systems, Asiaq – Greenland Survey, GEM (Greenland Ecosystem Monitoring) and ClimateBasis programme for providing meteorological observations, and Zackenberg Ecological Research Operations for facilitating logistics. This work forms part of the Nordic Centres of Excellence DEFROST and eSTICC and the EU FP7 project INTERACT.

**Figures**

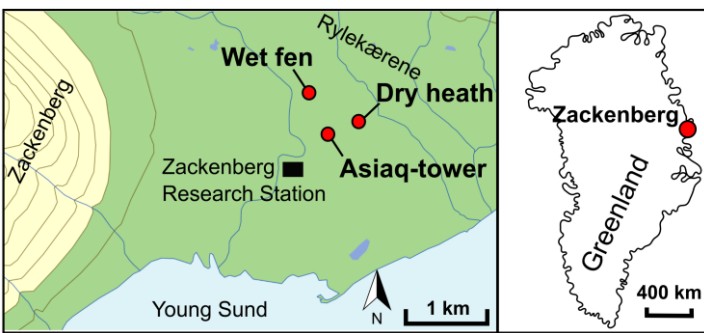

**Figure 1a:** Study location.

Location of the study sites in Zackenberg, Northeast Greenland (base map provided by NunaGIS).

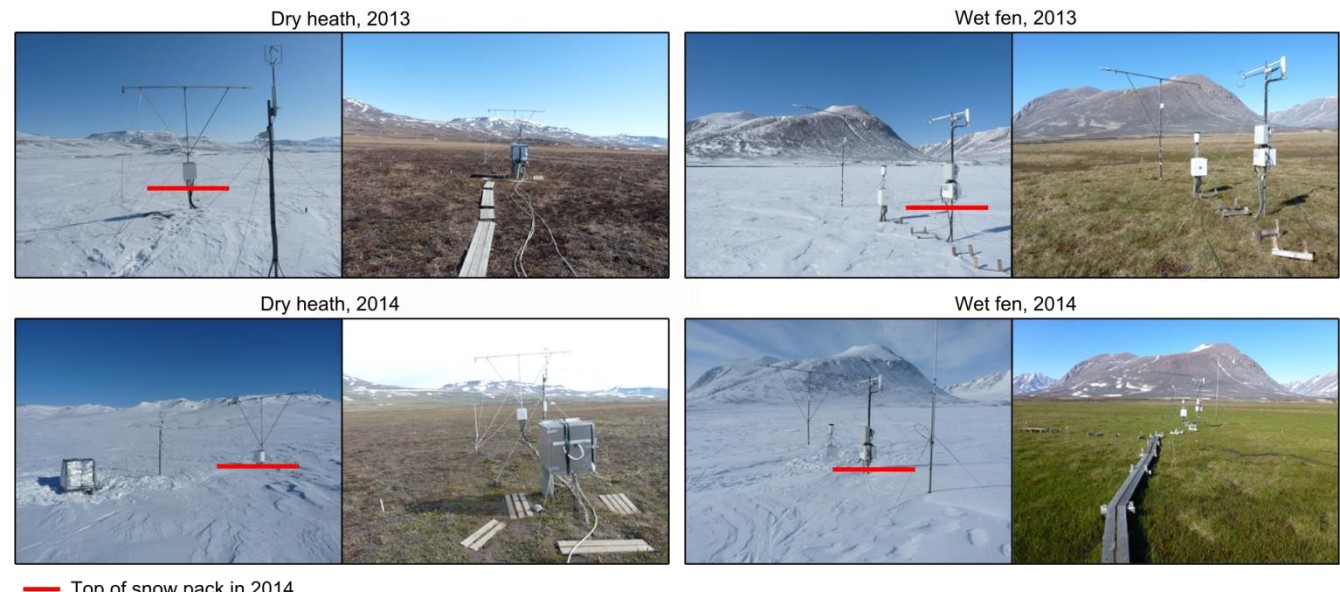

**Figure 1b:** Characteristics of the study sites.

Characteristics of the study sites during the beginning of the snow melt periods and during the growing seasons in 2013 and 2014.

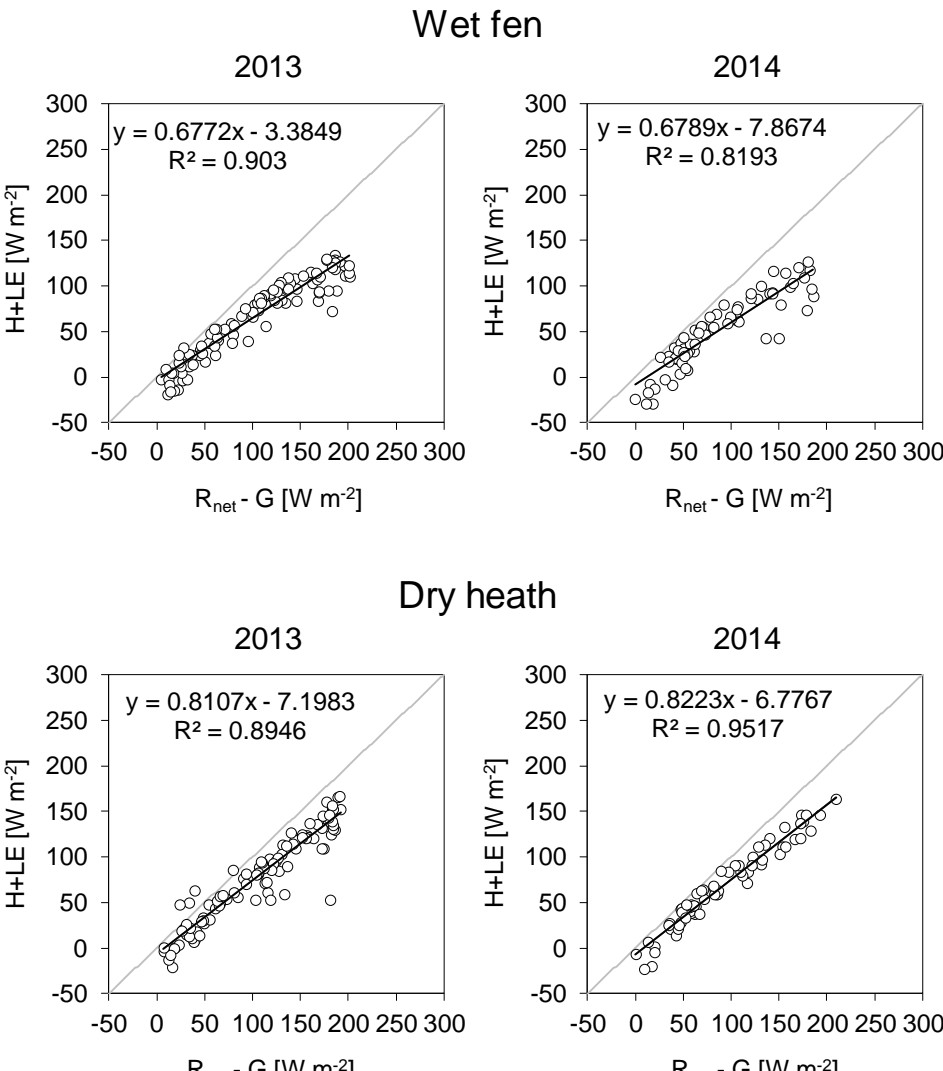

**Figure 2:** Energy balance closure.

Mean daily available energy ($R_{net} – G$) and turbulent heat fluxes ($H+LE$) at the wet fen (upper) and the dry heath (lower) during the observation periods in 2013 and 2014.

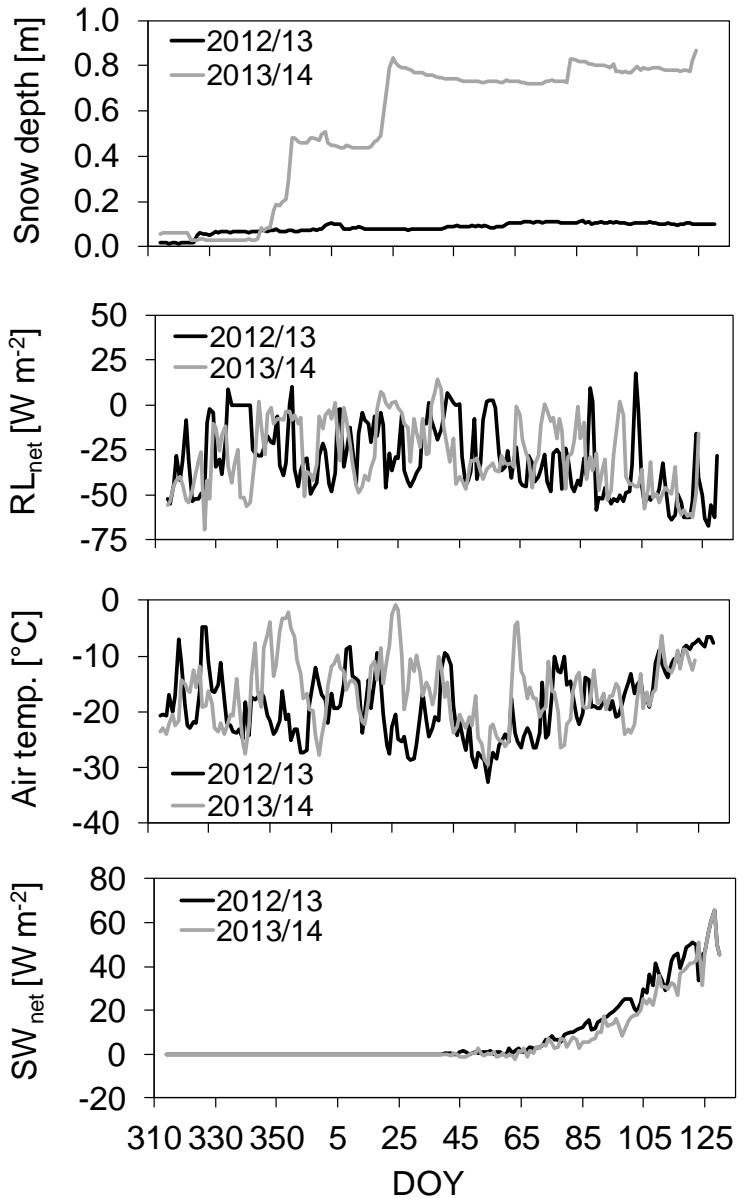

**Figure 3:** Polar night and pre-melt seasons of the winters 2012/13 and 2013/14.

5    Characteristics of snow cover development, air temperature, net longwave radiation ($RL_{net}$) and net shortwave radiation ($SW_{net}$) during the polar night periods and subsequent pre-melt seasons of the winters 2012/13 and 2013/14 (Asiaq-station).

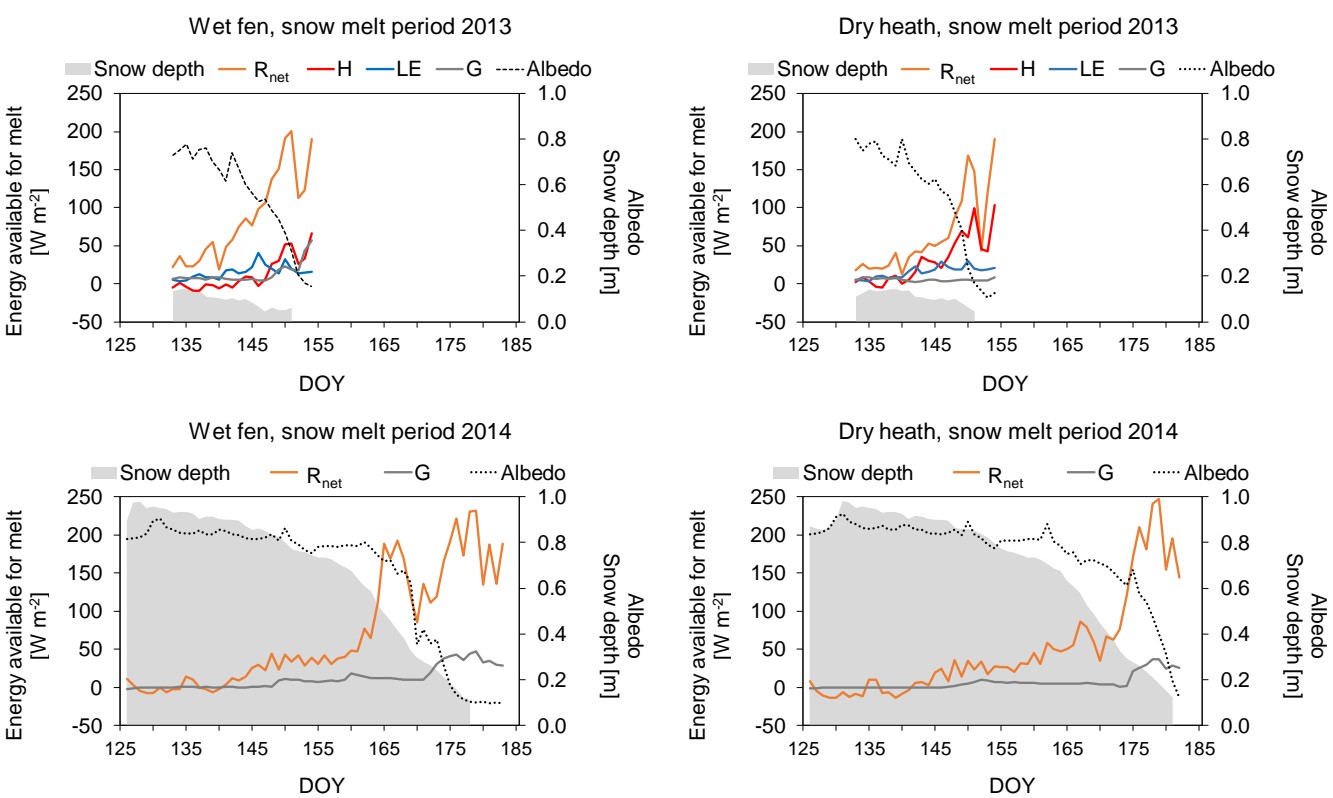

**Figure 4:** Snow melt periods 2013 and 2014.

Development of mean daily snow cover thickness, albedo, net radiation ($R_{net}$), sensible heat fluxes ($H$), latent heat fluxes ($LE$) and ground heat fluxes ($G$) at the wet fen (left) and dry heath (right) during the snow melt period in 2013 and 2014 and first 5 days of the growing season in 2013 and 2014.

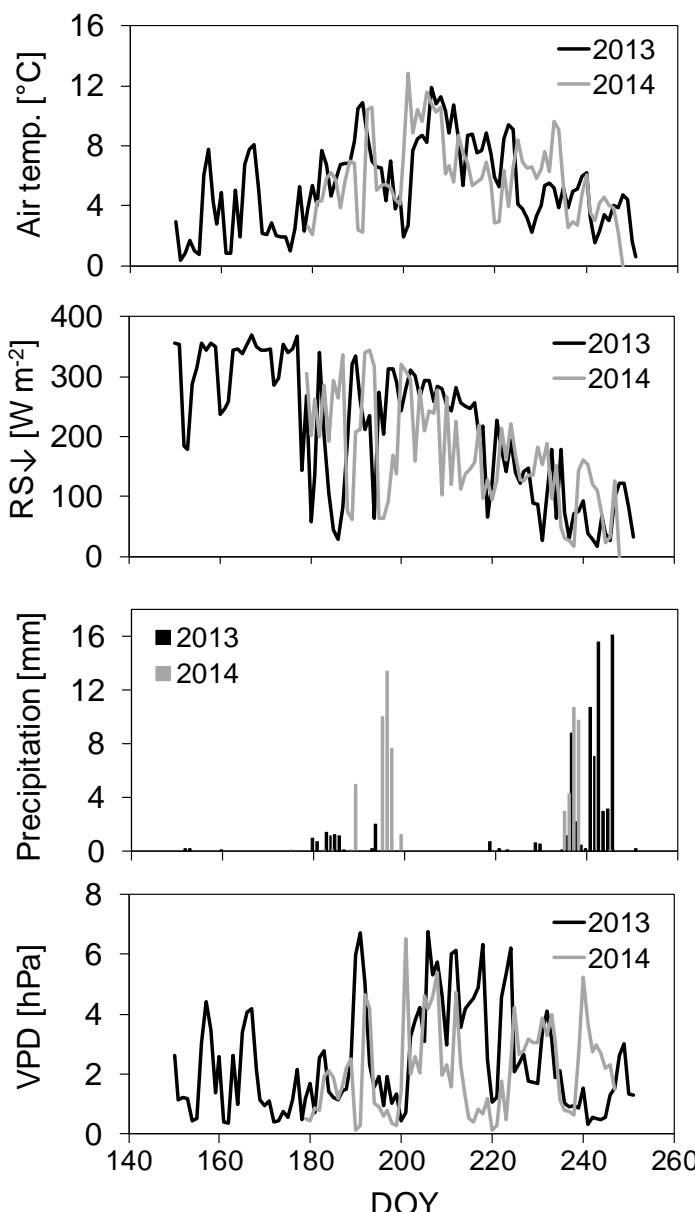

**Figure 5:** Growing seasons 2013 and 2014.

Development of mean daily air temperature, precipitation, incoming solar radiation (*RS↓*) and vapour pressure deficit (*VPD*) during the growing seasons in 2013 and 2014 (Asiaq-station).

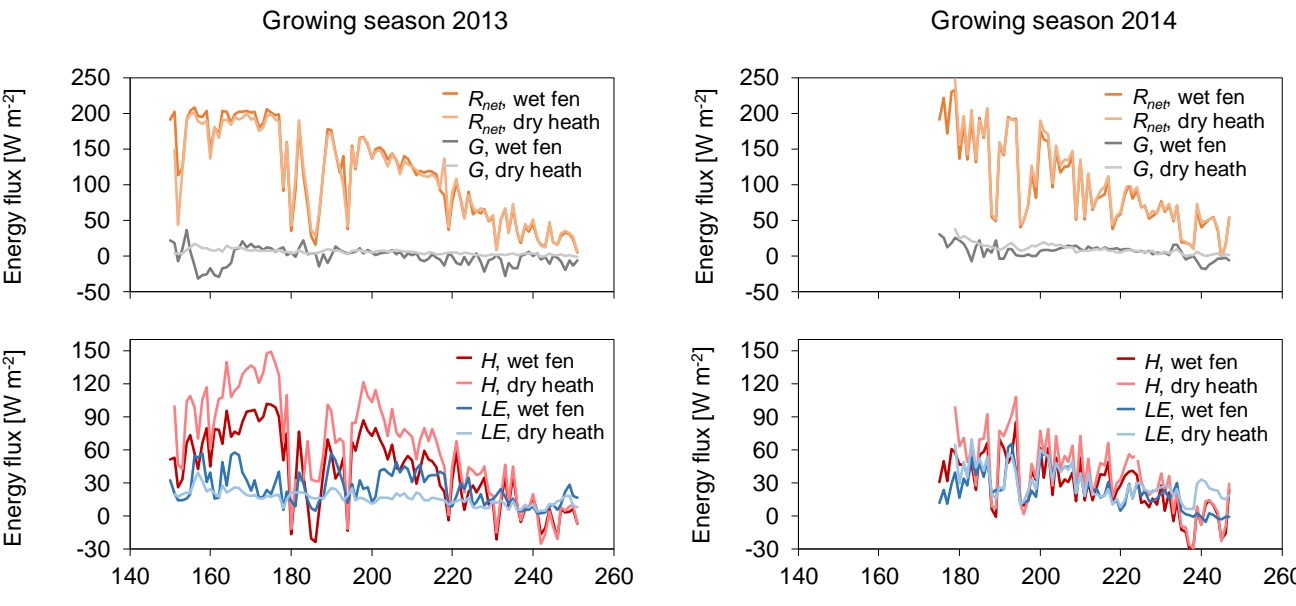

**Figure 6:** Growing season energy fluxes.

Mean daily net radiation ($R_{net}$), sensible heat flux ($H$), latent heat flux ($LE$) and ground heat flux ($G$) at the wet fen and dry heath during the growing seasons in 2013 (left) and 2014 (right).

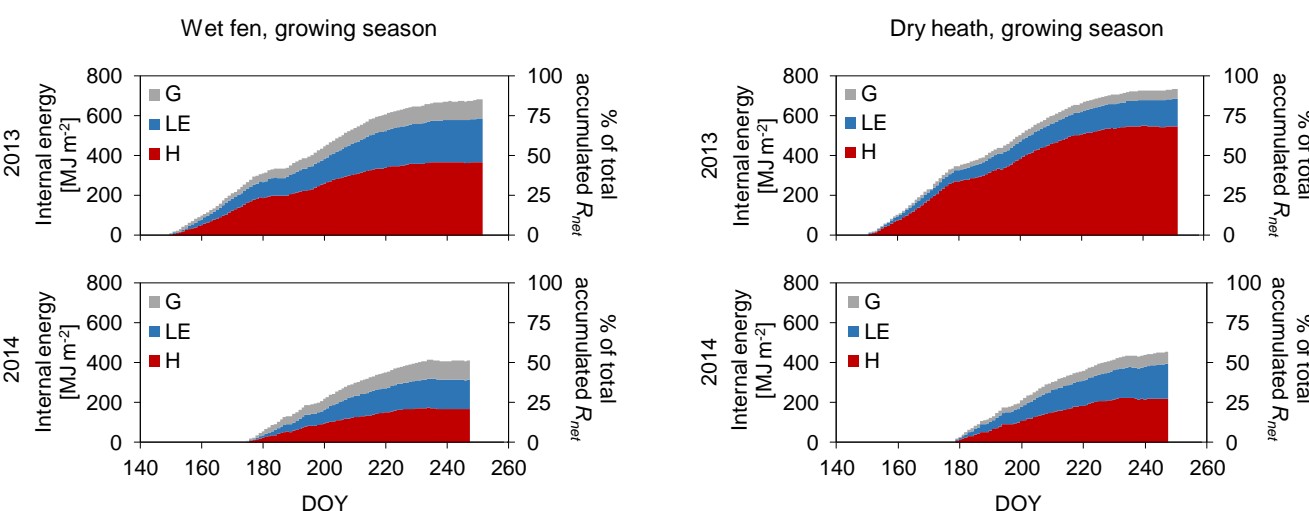

**Figure 7:** Development of internal energy.

Accumulated ground heat (*G*), latent heat (*LE*) and sensible heat (*H*) at the wet fen (left) and the dry heath (right) during the growing seasons in 2013 and 2014.

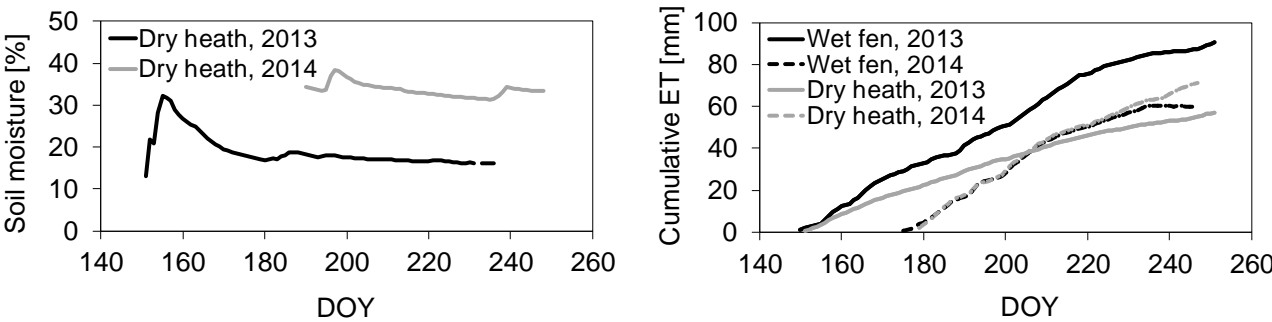

**Figure 8:** Development of soil moisture conditions (10 cm depth) and evapotranspiration (*ET*).

Mean daily soil moisture content at the dry heath (left) and cumulative evapotranspiration (*ET*) at the wet fen and dry heath during the growing season in 2013 and 2014 (right).

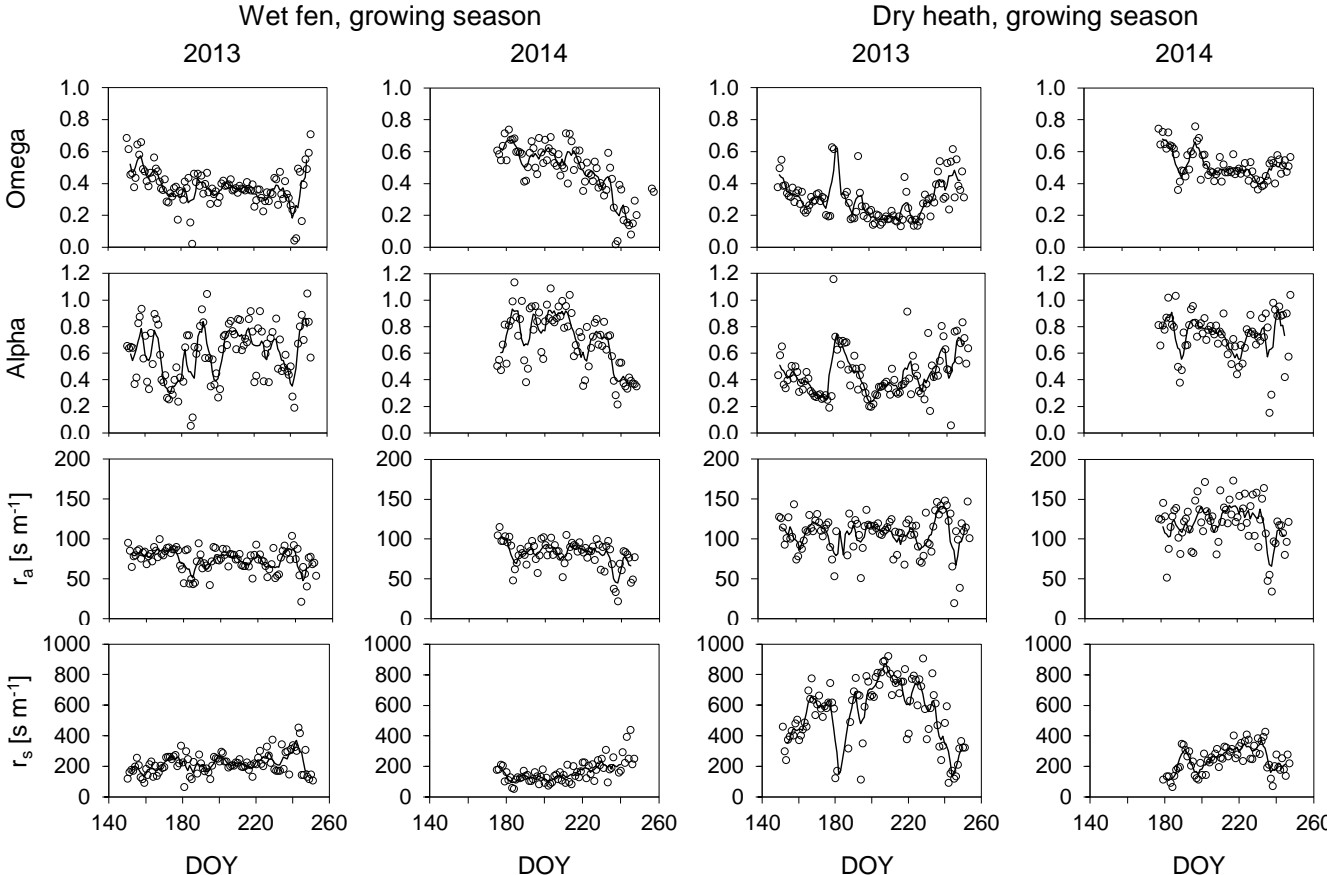

**Figure 9:** Omega, alpha, aerodynamic resistance and surface resistance.

Mean daily (empty circles) and 5-day running mean (solid line) of McNaughton & Jarvis decoupling factor (Omega), Priestley and Taylor Alpha-value (Alpha), aerodynamic resistance ($r_a$) and surface resistance ($r_s$) at the wet fen (left) and dry heath (right) during the growing seasons in 2013 and 2014.

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

**Tables**

**Table 1:** List of parameters and symbols.

| Name | Unit | Explanation | Location |
|------|------|-------------|----------|
| $RS{\downarrow}$ | W m$^{-2}$ | Incoming shortwave radiation | Wet fen, dry heath, Asiaq-tower |
| $RS{\uparrow}$ | W m$^{-2}$ | Outgoing shortwave radiation | Wet fen, dry heath, Asiaq-tower |
| $RL{\downarrow}$ | W m$^{-2}$ | Downwelling longwave radiation | Wet fen, dry heath, Asiaq-tower |
| $RL{\uparrow}$ | W m$^{-2}$ | Upwelling longwave radiation | Wet fen, dry heath, Asiaq-tower |
| $RS_{net}$ | W m$^{-2}$ | Net solar radiation [Eq. 2] | Wet fen, dry heath, Asiaq-tower |
| $RL_{net}$ | W m$^{-2}$ | Net longwave radiation [Eq. 2] | Wet fen, dry heath, Asiaq-tower |
| $R_{net}$ | W m$^{-2}$ | Net radiation [Eq. 2] | Wet fen, dry heath, Asiaq-tower |
| Albedo | | Surface albedo | Wet fen, dry heath, Asiaq-tower |
| $T_a$ | °C | Air temperature | Wet fen, dry heath, Asiaq-tower |
| $T_{surf.}$ | °C | Snow surface temperature | Wet fen, dry heath, Asiaq-tower |
| $T_s$ | °C | Soil temperature at 2 and 10 cm depth | Wet fen, dry heath |
| $H$ | W m$^{-2}$ | Sensible heat flux | Wet fen, dry heath |
| $LE$ | W m$^{-2}$ | Latent heat flux | Wet fen, dry heath |
| $G$ | W m$^{-2}$ | Ground heat flux | Wet fen, dry heath |
| $H/LE$ | | Bowen ratio | Wet fen, dry heath |
| Snow depth | m | Snow depth | Wet fen, dry heath, Asiaq-tower |
| Soil moisture | % | Soil moisture content | Wet fen, dry heath |
| Mixing ratio | g kg$^{-1}$ | Atmospheric mixing ratio | Wet fen, dry heath |
| $RH$ | % | Relative humidity | Wet fen, dry heath, Asiaq-tower |
| $VPD$ | hPa | Atmospheric vapour pressure deficit | Wet fen, dry heath, Asiaq-tower |
| $ET$ | mm d$^{-1}$ | Evapotranspiration | Wet fen, dry heath |
| Precip. | mm | Precipitation | Asiaq-tower |
| $\alpha$ | | Priestley-Taylor coefficient [Eq. 8] | Wet fen, dry heath |

| | | | |
|---|---|---|---|
| $u$ | m s$^{-1}$ | Wind speed | Wet fen, dry heath |
| $u_*$ | m s$^{-1}$ | Friction velocity | Wet fen, dry heath |
| $r_a$ | s m$^{-1}$ | Aerodynamic resistance [Eq. 5] | Wet fen, dry heath |
| $r_s$ | s m$^{-1}$ | Surface resistance [Eq. 6] | Wet fen, dry heath |
| $\Omega$ | | McNaughton & Jarvis Omega value [Eq. 7] | Wet fen, dry heath |
| NDVI | | Normalized Difference Vegetation Index | Wet fen |

**Table 2:** Summary of polar night and pre-melt season.

List of mean daily radiation components and environmental characteristics during the polar night seasons and pre-melt seasons in 2012/13 and 2013/14.

| | Polar night | | | | Pre-melt season | | | |
|---|---|---|---|---|---|---|---|---|
| | 2012/13 | | 2013/14 | | 2013 | | 2014 | |
| Duration | 10 Nov. – 4 Feb. | | 10 Nov. – 4 Feb. | | 5 Feb. – 12 May | | 5 Feb. – 5 May | |
| | Average (Standard error) | Range | Average (Standard error) | Range | Average (Standard error) | Range | Average (Standard error) | Range |
| $RS{\downarrow}$ (W m$^{-2}$) | 0 | 0 | 0 | 0 | 99.2 (8.6) | 0 – 586.9 | 87.0 (8.3) | 0 – 577.0 |
| $RS{\uparrow}$ (W m$^{-2}$) | 0 | 0 | 0 | 0 | -81.9 (6.8) | 0 – -449.5 | -76.4 (7.0) | 0 – -480.0 |
| $RL{\downarrow}$ (W m$^{-2}$) | 193.2 (3.9) | 133.4 – 293.9 | 213.1 (4.7) | 140.7 – 305.7 | 188.3 (3.1) | 142.1 – 269.8 | 194.6 (3.3) | 141.3 – 280.5 |
| RL↑ (W m$^{-2}$) | -219.2 (2.3) | -176.9 – -295.1 | -237.1 (3.1) | -190.4 – -307.7 | -224.1 (2.6) | -175.5 – -276.5 | -225.7 (2.5) | -173.3 – -285.8 |
| $RS_{net}$ (W m$^{-2}$) | 0 | 0 | 0 | 0 | 17.1 (1.9) | 0 – 65.8 | 11.0 (1.4) | 0 – 50.7 |
| $RL_{net}$ (W m$^{-2}$) | -27.0 (1.8) | -54.8 – 10.3 | -24.1 (2.1) | -69.6 – 7.7 | -35.4 (2.1) | -67. 4 – 17.5 | -31.1 (1.9) | -62. 7 – 14.2 |
| $R_{net}$ (W m$^{-2}$) | -27.0 (1.8) | -54.8 – 10.3 | -24.1 (2.1) | 69.6 – 7.7 | -18.0 (1.8) | -44.0 – 37.3 | -20.5 (1.7) | -48.2 – 15.3 |
| $T_a$ (°C) | -19.2 (0.6) | -28.7 – -4.8 | -15.2 (0.7) | -27.8 – -0.8 | -18.1 (0.7) | -23.5 – -6.5 | -17.6 (0.6) | -29.4 – -3.85 |
| $T_{surf.}$ (°C) | -24.1 (0.6) | -36.8 – -4.6 | -19.2 (0.8) | -32.4 – -1.7 | -22.8 (0.7) | -37.3 – -9.0 | -22.3 (0.7) | -38.0 – -6.7 |

| Air pressure (hPa) | 1010.0 (1.1) | 983.1 – 1030.6 | 1003.5 (1.2) | 956.9 – 1021.3 | 1019.1 (1.2) | 999.1 – 1050.0 | 1007.0 (1.1) | 975. 5 – 1030.4 |
|---|---|---|---|---|---|---|---|---|

**Table 3:** Summary of snow melt period.

List of mean daily surface energy balance and environmental characteristics during the snow melt period in 2013 and 2014 at the wet fen and dry heath site.

| | Wet fen, 2013 | | Wet fen, 2014 | | Dry heath, 2013 | | Dry heath, 2014 | |
|---|---|---|---|---|---|---|---|---|
| Duration | 13 May – 29 May | | 6 May – 23 June | | 13 May – 30 May | | 6 May – 27 June | |
| | Average (Standard error) | Range | Average (Standard error) | Range | Average (Standard error) | Range | Average (Standard error) | Range |
| $RS\downarrow$ (W m$^{-2}$) | 260.3 (19.3) | 111.0 – 355.9 | 292.4 (9.6) | 105.7 – 387.9 | 252.7 (17.6) | 113.4 – 340.6 | 290.5 (8.7) | 108.7 – 380.5 |
| $RS\uparrow$ (W m$^{-2}$) | -134.7 (9.4) | -68.9 – -204.6 | -205.7 (11.2) | -14.4 – -273.5 | -146.8 (11.1) | -44.2 – -213.0 | -213.2 (10.1) | -27.1 – -276.4 |
| $RL\downarrow$ (W m$^{-2}$) | 272.9 (5.6) | 237.5 – 314.6 | 275.3 (4.8) | 209.6 – 333.3 | 251.1 (6.3) | 213.7 – 295.6 | 276.9 (4.7) | 208.1 – 333.6 |
| $RL\uparrow$ (W m$^{-2}$) | -320.3 (2.3) | -309.5 – -343.3 | -314.1 (3.1) | -264.3 – -343.4 | -302.9 (2.3) | -293.5 – -328.5 | -316.1 (3.1) | -263.5 – -363.2 |
| $RS_{net}$ (W m$^{-2}$) | 125.5 (18.5) | 29.7 – 283.8 | 86.7 (7.8) | 18.6 – 248.9 | 105.9 (15.9) | 24.5 – 259.0 | 77.3 (7.9) | 15.1 – 321.5 |
| $RL_{net}$ (W m$^{-2}$) | -47.4 (6.4) | -83.3 - -6.6 | -38.8 (2.6) | -69.6 - -2.9 | -51.7 (7.0) | -97.4 - -7.3 | -39.2 (2.8) | -80.2 - -1.9 |
| $R_{net}$ (W m$^{-2}$) | 78.1 (13.0) | 18.6 – 200.5 | 47.9 (8.2) | -8.1 – 192.6 | 54.2 (10.1) | 12.8 – 168.2 | 38.1 (7.7) | -14.3 – 241.4 |
| Albedo | 0.57 (0.03) | 0.31 – 0.78 | 0.68 (0.02) | 00.26 – 0.90 | 0.63 (0.04) | 0.17 – 0.80 | 0.73 (0.01) | 0.47 – 0.92 |
| $H$ (W m$^{-2}$) | 7.6 (4.3) | -9.1 – 52.8 | - | - | 25.0 (6.5) | -4.8 – 99.5 | - | - |
| $LE$ (W m$^{-2}$) | 15.6 (2.2) | 3.3 – 40.2 | - | - | 14.9 (1.8) | 3.5 – 30.2 | - | - |

| | | | | | | | | |
|---|---|---|---|---|---|---|---|---|
| $G$ (W m$^{-2}$) | 8.4 (1.2) | 4.0 – 23.0 | 7.2 (1.2) | -2.3 – 38.3 | 5.1 (0.4) | 1.7 – 8.5 | 5.1 (1.1) | -1.8 – 37.1 |
| $T_a$ (°C) | -1.6 (0.4) | -3.7 – 3.0 | -1.4 (0.7) | -10.7 – 9.6 | -1.6 (0.4) | -3.7 – 3.0 | -1.2 (0.6) | -10.7 – 9.6 |
| $T_s$ (°C) [1] | -6.8 (0.6) | -10.5 - -0.3 | -6.3 (0.6) | -11.8 – 1.7 | -2.4 (0.5) | -7.7 – 0-3 | -4.8 (0.7) | -11.4 – 3.5 |
| Air pressure (hPa) | 1017.0 (1.0) | 1011.3 – 1025.3 | 1014.9 (0.9) | 1000.4 – 1028.0 | 1017.0 (1.0) | 1011.3 – 1025.3 | 1014.5 (0.9) | 1000.4 – 1028.0 |
| $ET$ | 0.51 (0.08) | 0.12 – 1.41 | - | - | 0.52 (0.07) | 0.13 – 1.06 | - | - |

[1]Measured at 2 cm depth

**Table 4:** Summary of growing season.

List of mean daily surface energy balance and environmental characteristics during the growing seasons in 2013 and 2014 at the wet fen and dry heath site.

| | Wet fen | | | | Dry heath | | | |
|---|---|---|---|---|---|---|---|---|
| | 2013 | | 2014 | | 2013 | | 2014 | |
| Duration | 30 May – 8 Sept. | | 24 June – 4 Sept. | | 31 May – 8 Sept. | | 28 June – 4 Sept. | |
| | Average (Standard error) | Range | Average (Standard error) | Range | Average (Standard error) | Range | Average (Standard error) | Range |
| $RS\downarrow$ (W m$^{-2}$) | 211.5 (10.9) | 17.7 – 369.3 | 179.0 (10.8) | 17.4 – 375.1 | 210.4 (10.9) | 15.7 – 369.6 | 171.2 (10.7) | 15.7 – 370.6 |
| $RL\uparrow$ (W m$^{-2}$) | -361.7 (1.4) | -329.2 – -390.6 | -364.4 (1.6) | -340.1 – -398.2 | -356.0 (1.5) | -319.4 – -381.7 | -366.2 (1.8) | -340.3 – -399.9 |
| $RL\downarrow$ (W m$^{-2}$) | 305.1 (2.2) | 257.4 – 350.9 | 314.0 (2.5) | 26.08 – 357.0 | 291.5 (2.8) | 226.5 – 348.6 | 313.6 (2.6) | 261.4 – 356.5 |
| $RS\uparrow$ (W m$^{-2}$) | -41.1 (2.2) | -2.1 – -83.1 | -28.3 (1.8) | -2.1 – -56.9 | -40.7 (2.2) | -2.1 – -74.9 | -19.7 (1.3) | -0.7 – -43.9 |
| $RS_{net}$ (W m$^{-2}$) | 170.4 (8.7) | 15.7 – 298.3 | 150.7 (9.1) | 15.3 – 322.2 | 169.7 (8.8) | 13.6 – 298.3 | 151.5 (9.4) | 15.0 – 336.4 |
| $RL_{net}$ (W m$^{-2}$) | -56.6 (2.8) | -98.4 – -2.4 | -50.5 (3.1) | -98.4 – -4.5 | -71.4 (3.5) | -111.9 – -4.4 | -52.6 (3.4) | -104.8 – -4.5 |
| $R_{net}$ (W m$^{-2}$) | 114.2 (6.4) | 5.2 – 208.3 | 100.2 (7.1) | -2.0 – 231.9 | 111.4 (6.1) | 6.3 – 201.3 | 98.9 (7.0) | 0.3 – 246.8 |
| Albedo | 0.20 (0.003) | 0.13 – 0.31 | 0.17 (0.005) | 0.07 – 0.28 | 0.16 (0.004) | 0.08 – 0.32 | 0.13 (0.006) | 0.06 – 0.26 |
| $H$ (W m$^{-2}$) | 41.5 (3.4) | -23.8 – 101.7 | 26.6 (2.9) | -29.5 – 85.1 | 62.6 (4.6) | -25.9 – 149.5 | 36.9 (3.8) | -29.6 – 107.7 |

| Variable | Mean (SE) | Range | Mean (SE) | Range | Mean (SE) | Range | Mean (SE) | Range |
|---|---|---|---|---|---|---|---|---|
| $LE$ (W m$^{-2}$) | 25.2 (1.5) | 1.8 – 57.2 | 23.3 (2.0) | -5.9 – 65.7 | 16.0 (0.6) | 3.6 – 39.1 | 29.3 (1.8) | 5.9 – 69.0 |
| $G$ (W m$^{-2}$) | 10.8 (0.9) | -3.9 – 56.5 | 15.2 (1.3) | -2.1 – 47.5 | 5.6 (0.3) | -1.5 – 16.9 | 10.1 (0.9) | -0.4 – 37.3 |
| $H/LE$ | 1.6 (0.1) | 0.16 – 4.3 | 1.1 (0.1) | 0.28 – 3.09 | 4.3 (0.2) | 0.03 – 8.3 | 1.6 (0.1) | 0.28 – 3.09 |
| $H/R_{net}$ | 0.36 (0.01) | 0.03 – 0.57 | 0.27 (0.01) | 0.01 – 0.40 | 0.54 (0.02) | 0.11 – 0.88 | 0.38 (0.01) | 0.05 – 0.56 |
| $LE/R_{net}$ | 0.22 (0.01) | 0.04 – 0.61 | 0.23 (0.01) | 0.01 – 0.35 | 0.18 (0.01) | 0.08 – 0.56 | 0.33 (0.01) | 0.12 – 0.70 |
| $G/R_{net}$ | 0.12 (0.01) | 0.04 – 0.61 | 0.17 (0.01) | 0.01 – 0.35 | 0.05 (0.003) | 0.07 – 0.56 | 0.10 (0.01) | 0.12 – 0.70 |
| $T_a$ (°C) | 5.3 (0.3) | 0.4 – 11.9 | 5.8 (0.3) | 0.5 – 12.9 | 5.3 (0.3) | 0.4 – 11.9 | 6.1 (0.3) | 2.1 – 12.9 |
| $T_s$ (°C) [1] | 4.0 (0.3) | -2.0 – 8.3 | 5.0 (0.2) | -0.6 – 9.6 | - | - | 3.3 (0.2) | -0.5 – 6.5 |
| Soil moisture (%) | - | - | - | - | 19.0 (0.4) | 13.1 – 32.2 | 33.6 (0.2) | 31.4 – 38.3 |
| Air pressure (hPa) | 1004.4 (0.7) | 989.4 – 1018.3 | 1009.9 (0.6) | 999.6 – 1019.3 | 1004.3 (0.7) | 989.4 – 1018.3 | 1009.9 (0.6) | 999.6 – 1019.3 |
| Mixing ratio (g kg$^{-1}$) | 6.6 (0.2) | 4.1 – 10.9 | 6.5 (0.2) | 4.2 – 11.2 | 6.6 (0.2) | 4.1 – 10.9 | 6.6 (0.2) | 4.6 – 11.2 |
| $RH$ (%) | 71.1 (1.6) | 33.9 – 97.2 | 76.5 (1.8) | 42.4 – 98.6 | 71.2 (1.6) | 33.9 – 97.1 | 75.5 (1.9) | 42.4 – 98.6 |
| $VPD$ (hPa) | 2.4 (0.2) | 0.3 – 6.7 | 2.0 (0.2) | 0.1 – 6.5 | 2.4 (0.2) | 0.3 – 6.7 | 2.2 (0.2) | 0.1 – 6.5 |
| $ET$ (mm d$^{-1}$) | 0.89 (0.05) | 0.06 – 2.0 | 0.82 (0.07) | -0.21 – 2.31 | 0.56 (0.02) | 0.13 – 1.38 | 1.03 (0.06) | 0.21 – 2.43 |

| | | | | | | | | |
|---|---|---|---|---|---|---|---|---|
| $\alpha$ | 0.60 (0.02) | 0.06 – 1.05 | 0.69 (0.03) | 0.22 – 1.14 | 0.44 (0.02) | 0.06 – 1.16 | 0.74 (0.02) | 0.15 – 1.04 |
| $u$ (m s$^{-1}$) | 2.7 (0.1) | 1.4 – 11.0 | 2.4 (0.2) | 1.0 – 9.9 | 2.8 (0.2) | 1.4 – 12.4 | 2.5 (0.2) | 1.0 – 10.3 |
| $u_*$ (m s$^{-1}$) | 0.25 (0.01) | 0.12 – 0.91 | 0.22 (0.02) | 0.09 – 0.95 | 0.24 (0.02) | 0.09 – 1.12 | 0.20 (0.01) | 0.09 – 0.80 |
| $r_a$ (s m$^{-1}$) | 74.2 (0.5) | 4.8 – 159.6 | 81.5 (0.7) | 5.2 – 159.4 | 107.4 (1.0) | 4.4 – 337.3 | 119.7 (1.3) | 7.2 – 336.7 |
| $r_s$ (s m$^{-1}$) | 217.7 (7.0) | 65.5 – 455.0 | 165.4 (8.4) | 52.5 – 441.5 | 555.7 (21.9) | 93.2 – 922.9 | 246.7 (10.8) | 64.2 – 428.1 |
| $\Omega$ | 0.38 (0.01) | 0.02 – 0.71 | 0.48 (0.02) | 0.01 – 0.74 | 0.30 (0.01) | 0.14 – 0.63 | 0.52 (0.01) | 0.36 – 0.76 |
| NDVI | 0.45 (0.004) | 0.01 – 0.68 | 0.49 (0.006) | 0.03 – 0.74) | - | - | - | - |

[1]Measured at 10 cm depth