# Peer review of "Two years with extreme and little snowfall: Effects on energy partitioning and surface energy exchange in a high-Arctic tundra ecosystem"

_The Cryosphere, 2016_

## Referee Comment (RC1) · J. McFadden (Referee) · 21 Apr 2016

This paper reports results from a 2-year field measurement campaign at Zackenberg, Greenland to measure seasonal time series of energy budget components in two contrasting arctic vegetation types, wet fen and dry heath tundra. The paper makes use of the fact that there was a strong contrast in snow cover between the two years to infer what changes in surface energy exchange might be expected under a warming (and higher precipitation) arctic climate. Changes in the arctic energy budget, especially those studied seasonally, are important. The variety of ecosystem types that have been measured–including in Greenland–is relatively few, making this an important contribution to our knowledge of cold land processes. The manuscript is very clearly written,

well organized, and the data are well presented (but see a few comments below). The conclusions are reasonable and are supported by the data. I recommend the paper for publication with only minor revisions. In my view, the paper is very good and the detailed comments below are given as constructive suggestions to improve the final version.

General comments:

1) Figure design. The figures are very nice and clearly labeled; however, some of them are confusing to read. For example, in Figs 3a and 3b the stacked bar plots make it very hard to see the differences between sites and years. Stacked bars are difficult in general, but the large difference in the size of the trends (e.g. the tiny negative values of Rnet) make it really difficult to discern, especially in these small panels. I think the data will come across to the reader better and the paper will have better impact if you can find another way to show these data. Maybe it would be a non-stacked bar plot, a line plot, or something else. Maybe they need to be made larger so that you show only one vegetation type at a time. I understand the reasons why you laid out the panels as you did, it's just that they end up difficult to read and so the result will be better if you can revise them to show the various data values more clearly.

Secondly, the use of color is not always allowing the reader to quickly pick out what is what. Sometimes the colors are inconsistent, for example in Fig 5, H and LE are shown in red and blue, and G is in green. But in Fig 6, H and LE follow the same red and blue style, but G is now in gray. Earlier in the paper (Figs 3a and 3b), the same shades of red and blue are used instead for albedo and snow depth. I think you might be better off to remove color from most of the plots where it is possible, and reserve the use of color only where it is the best/only way to show the differences. For example, you could symbolize albedo and snow depth with black lines of different thicknesses, or using dashes. I would suggest placing all of the color figures all together out on a table and then deciding on a system so that the same type of color or symbols are used consistently, and simplify them as much as you can by not using color at all unless it is

needed.

2) References. The paper does a good job of citing relevant literature overall, but its impact would be improved by making more direct comparisons to other arctic energy budget studies, especially those outside of Greenland. There are still few enough such studies that this is valuable to the scientific community to provide a bigger context for your findings. The manuscript did cite the big review paper of Eugster et al. (2000), but you didn't use that paper to actually compare the energy budget patterns you found to the patterns reported for other arctic ecosystems. At least in the North American Arctic where I have worked, there are other studies that have measured the same energy budget components and even looked at some of the same wet-dry ecosystem comparisons as you are making here. I think the paper would be stronger if it linked into the Arctic literature a little better by directly stating whether the results obtained here are consistent with or different from the values and the overall patterns found in a broader set of arctic sites. This need not be a major addition to the text, but just digging in a bit deeper for each such value or pattern that you highlight in the Discussion.

Detailed comments:

Abstract. The last part of the final sentence in the abstract is rather vague. Simply "increasing interannual variability" seems rather general and less interesting that what you actually found. I suggest thinking about how your could "sharpen" this final statement to make it more specific and impactful. I could imagine you might want to say something about how interannual climate variability (winter precip) has been shown to have different feed backs to surface energy partitioning depending on ecosystem type...

p 4, Measurements. Accepted that you are using standard CarboEurope type procedures, but can you add a citation to a paper from your flux site that explains site-specific procedures and conditions? If not, then can you please add just a brief description of key site specific information such as how data were screened (spikes, low turbulence,

unfavorable meteorological conditions) and how many gaps (what percentage of the record, what percentage of the daytime values that went into the energy budget measurements, or whatever you think is relevant)? I see that you later explain that you used the MPI online gap-filling tool to do the gap filling. I am just requesting a little info to characterize data screening and the overall situation with gaps at the two towers. And any other key, site-specific information or procedures that you think are important to the reader. If all of this is summarized in a different flux paper, then it is fine to simply cite it as above, and say that those details are provided in the cited paper.

p 7, line 10-11: It is very difficult to see the negative values of Rnet in the figure because of its vertical scale and the bar type. Please see general comment above on the figures.

p 8, line 6ff: Please provide a short sentence or two explaining how the 2 study years compare to the long-term climate of the site. Which of the study years was more typical of "mean" climate conditions for this location in terms of the different seasons and meteorological variables? Which aspects were most atypical—only the low-snowfall year, or anything else?

p 12, line 4: Recommend changing "shields off" to "reflects".

p 13, Energy Budget Closure section: I suggest that you move the energy budget closure discussion somewhere further up in the Discussion section, rather than as the last item. There are two reasons for this. First, it makes the paper less impactful if the very last item discussed is a technical system-performance assessment like energy budget closure, rather than one of the main findings on your scientific questions. Second, it is arguable that the reader should know whether you think the energy closure imbalance is typical of arctic sites or is in any way a problem before reading onward to the main scientific findings. You might even put it first in the discussion, or at least somewhat closer to the begining of the discussion, and not the very last item.

In addition, do you think that variations in the depth of permafrost may have been

a source of uncertainty/error in estimating the ground heat flux and energy budget closure, and if yes, you could mention that. Finally, I suggest that you end the part about energy budget closure with a comment on how the lack of closure would (or would not) affect your findings. For example, you may believe that the H and LE fluxes are OK and that the closure error is mostly in G. Rn is not affected, and so forth. I think it would make the paper stronger if you could provide a short sentence or 2 at most that interprets what the closure error means for the results you have obtained here.

p 14, line 8: Change "discusses" to "discussed"

Some other arctic energy budget references (not exhaustive, just some close examples to what you have done here)

Sturm, M., et al. (2005). "Changing snow and shrub conditions affect albedo with global implications." Journal of Geophysical Research-Biogeosciences 110(G01004).

McFadden, J. P., et al. (2003). "A regional study of the controls on water vapor and CO2 exchange in arctic tundra." Ecology 84(10): 2762-2776.

McFadden, J. P., et al. (1998). "Subgrid-scale variability in the surface energy balance of arctic tundra." Journal of Geophysical Research 103(D22): 28947–28961.

Rouse, W. R. (2000). "The energy and water balance of high-latitude wetlands: controls and extrapolation." Global Change Biology 6: 59–68.

Lafleur, P. M. and W. R. Rouse (1988). "The influence of surface cover and climate on energy partitioning and evaporation in a subarctic wetland." Boundary-Layer Meteorology 44(4): 327–348.

Lafleur, P. M., et al. (1987). "Components of the surface radiation balance of subarctic wetland terrain units during the snow-free season." Arctic, Antarctic, and Alpine Research 19(1): 53–63.

---

## Referee Comment (RC2) · Anonymous Referee #2 · 25 Apr 2016

In this study the authors assess the impact of strong inter-annual variability in snow accumulation during two subsequent years (2013, 2014) on the land-atmosphere interactions and surface energy exchange in two well instrumented high-Arctic tundra ecosystems under different moisture regimes (wet fen and dry heath) in Zackenberg, Northeast Greenland. The study takes advantage of the natural laboratory conditions of strongly different snowcover regimes between the two years, which motivates this study.

Their results suggest that in a changing climate with higher temperature and more precipitation the surface energy balance of this high-Arctic tundra ecosystem may experience a further increase in the inter-annual variability of energy accumulation, par-

titioning and redistribution.

I think the experimental setup is nice and clear with two differing ground moisture regimes being complemented by two strongly different snowcover regimes. In addition the paper is well organised and clear with precise method descriptions and a clear analysis. The paper is quite straightforward, primarily describing and interpreting the meteorological measurements in the context of their experimental setup and through seasonal changes (polar night, snowmelt, growing season). Due to the paucity of such measurements in the high Arctic this study is an important contribution to knowledge about atmosphere-surface dynamics in the high Arctic and recommend publishing subject to a few minor comments below.

COMMENTS

1. p.2 l.28– You need to mention the two site setup early in this paragraph as you just drop 'at our two high-Arctic sites' in at the end rather unexpectedly.

2. P.4 l.6 : this snow depth measurement comes from the Asiaq station or is made directly at the tower? If at the tower what's the instrument? If at the Asiaq station can you comment on representativeness?

3. p.4 l.14 can you mention the soil depths you measured at?

4. I wasn't able to identify which model OTT pluvio you used based on the reference (p.4 l.17). 5. How do you power your stations? Particularly during the polar night?

6. Perhaps a parameter table would be a useful look up for this paper with categories of: units, measured/derived, location, instrument (if measured), temporal resolution etc.

7. I found the last paragraph of Methods (p.6 l.1-10) where you define the "seasons": polar night, melt and growing a little disconnected. Obviously, you organise your results according to these categories which I think is nice, but you could add to this description that this is how you will present the data and why this is informative. This would make

the 'story' flow a little better.

8. I feel like the conclusions are missing some kind of outlook to what next ie. integrating models to scale results/ investigate other aspects of the enrgy balance or strategies to reduce the energy balance closure problem. I think a few sentences reflecting on ways of building on this study with further work would be useful.

9. Figure 1a: can you mark the Asiaq station on the map?

10. Figure 2: mention which site this is in the caption.

11. Picking up on the comments of J . McFadden and while I agree the stacked plots (Figure 3a/b) make it tricky to identify trends in individual years - I think the key point the authors intend to show is the cumulative energy inputs from all components. If that's the intention I would say some form of cumulative presentation is important.

12. Not immediately obvious which lines the axis refer to in Figure 6b - soil moisture is indicated on the black line, perhaps can do the same with blue/red lines (evaporation?).

---

## Referee Comment (RC3) · A. J. Dolman (Referee) · 11 May 2016

This is useful contribution to the arctic surface flux observation literature as the study documents a nice case study of control of the amount of snow (snow depth) on the subsequent evolution of the turbulent surface fluxes and melt. As such, and also given the amount of work it involves to get the data in such an environment, it deserves to be published. I do have some comments however, some major, some minor, that I hope may improve the paper.

Main comments.

First. The paper contains an awful lot of numbers, and no error estimates at all. In the

table the SD is given, but if that would be a good measure (and we know it is not), most of the data would fall within the same probability distribution. My first suggestion would be to abandon the use of SD and give the range, and give an error estimate of all your measurements. That helps to assess the significance of your difference.

Second. The energy balance closure is vital to the whole exercise in calculating the amount of energy available for melt as a residual from the energy balance. While I agree that a considerable amount of variability is expected, a value of 67% is very low and needs a little more explaining that referring to site heterogeneity. To show the validity of the eddy covariance measurements I would suggest to include a spectral analysis of the measurements. A good co-spectrum adds to the reliability and acceptance of the data. I also suggest to include this analysis in the description of the methodology, where it belongs, and not include as as an afterthought in the discussion.

Third. Soil moisture. In terms of controlling factors, soil moisture is the key term that controls the partitioning of the energy budget terms. Precipitation and snow depth are just proxies in that sense. I am surprised that only in Figure 6 soil moisture is used. In fact how it is measured is not mentioned at all in $ 2.2. Was it only measured at the dry heath. Please explain and use the data!

Fourth. Overall the analysis is very descriptive, even lacklustre at times. This is a pity as the data are very valuable! For example when parameters like surface resistance of omega are calculated there is no real effort to explain or interpret these (I find the big difference between the yearly wet fen values somewhat worrying though, given the magnitude of the difference; even between different vegetation types you would not expect such a big difference). This really needs some work. For instance if a wetter Arctic would imply less H, would that provide a negative feedback on the warming trend? There are plenty of such questions to ask given the data and I encourage the authors to think these through and by doing so add more meat to the discussion. I am also surprised to find that there is no mention of the Kasurinen et al., 2014, GCB study at all, given some of the co-authors participated in that study. This does provide a very

useful benchmark for the present study.

Fith. I would rephrase the title to make it a better aligned with the content. Something like a "A comparison of surface energy budgets of … in two years with extreme and little snowfall".

Minor comments

P1 l10. The use of interannual suggest that many years are used. I would suggest to not use this word and stick to the comparison of a snow rich and snow poor year (see above remark on title as well). P3 l27-30. This should be part of the introduction, not site description. P4 $2.2 Measurements. Have the systems be run side by side in a comparison experiment to show that they provide the same fluxes when at the same site? What are otherwise the errors you expect in the fluxes? P5 l11-13. Is there a way you can quantity the error that this would generate on your estimates of energy available for melt. P6 l28-30. I would guess that synoptic variability and weather dynamics also play a role here. You are expressing an extreme 1-D view of the atmosphere here. P7 28-29. This is an example of my first major comment. Are these values really different, or do they fall within you measurement error (given your energy closure for instance)? P8 l16 gives an another example. P8 l26-27. Is there any way you can relate this to greenness, density of vegetation as well, or is this really just an effect of the relative contribution of snow versus vegetation. I am asking also in relation to Fig 1, where a different colour seems to be visible for the different years. p9. $3.3.4. This part really needs some more work and check on the values of Rs in the dry heat growing season. Also it may be better to define a period of maximum gs, rather that show the average which is biased by the shoulder values of the season. Reference here also Kasurinen et al, 2014. This part is presently pretty shallow, I am afraid to say. P12 l27. I guess you mean soil moisture rather than groundwater? Otherwise how did you determine this? Figure 3a and 3b. Can you add the snowmelt of 3b just to 3a? This avoids repetition of the albedo and snow depth and temperature plot. You adjust the time period to the longer one of 3b. Figure 5. Can you reduce the scale of the
Y-axis in the lower panel plots so that differences are more visible? Figure 6. Can you add a second y-axis that gives the % of use of available energy for the different fluxes? Figure 6b is confusing as it does not fit with 6a (it shows both the dry tundra and wet fen and both years) and the structure of all the other plots. Make it a separate plot.

———————————————

---

## Author Comment (AC1) · 8 Jun 2016

In this study the authors assess the impact of strong inter-annual variability in snow accumulation during two subsequent years (2013, 2014) on the land-atmosphere in-teractions and surface energy exchange in two well instrumented high-Arctic tundra ecosystems under different moisture regimes (wet fen and dry heath) in Zackenberg, Northeast Greenland. The study takes advantage of the natural laboratory conditions of strongly different snowcover regimes between the two years, which motivates this study.

Their results suggest that in a changing climate with higher temperature and more precipitation the surface energy balance of this high-Arctic tundra ecosystem may ex-perience a further increase in the inter-annual variability of energy accumulation, partitioning and redistribution.

I think the experimental setup is nice and clear with two differing ground moisture regimes being complemented by two strongly different snowcover regimes. In addi-tion the paper is well organised and clear with precise method descriptions and a clear analysis. The paper is quite straightforward, primarily describing and interpreting the meteorological measurements in the context of their experimental setup and through seasonal changes (polar night, snowmelt, growing season). Due to the paucity of such measurements in the high Arctic this study is an important contribution to knowledge about atmosphere-surface dynamics in the high Arctic and recommend publishing sub-ject to a few minor comments below.

**COMMENTS**

1. p.2 l.28– You need to mention the two site setup early in this paragraph as you just drop 'at our two high-Arctic sites' in at the end rather unexpectedly.
   - *The paragraph was updated according to the reviewer's comment (p. 2 l. 19).*

2. P.4 l.6 : this snow depth measurement comes from the Asiaq station or is made directly at the tower? If at the tower what's the instrument? If at the Asiaq station can you comment on representativeness?
   - *Snow depth measurements were conducted at both sites (fen and heath), using snow depth sensors (SR50A, Campbell Scientific, USA). The structure of the paragraph was changed updated with information on snow depth sensor (p. 4, l. 12-16).*

3. p.4 l.14 can you mention the soil depths you measured at?
   - *The paragraph was updated with information on measurement depths of the soil temperature (2, 10, 20, 40, 50 and 60 cm), soil heat flux (4 cm depth) and net radiation (3 m height) (p. 4, l. 12-16).*

4. I wasn't able to identify which model OTT pluvio you used based on the reference (p.4 l.17).

- *The sensor model was updated in the manuscript (52203, R. Young Company, UK) (p. 4, l. 18).*

5. 5. How do you power your stations? Particularly during the polar night?

- *Power supply for all stations was provided by diesel generators from the nearby Zackenberg Research Station (May to October), and solar panels and a wind mill (Superwind 350, superwind GmbH, Germany) during the period when the research station was closed. The information was added to the manuscript (p. 4, l. 21-22).*

6. Perhaps a parameter table would be a useful look up for this paper with categories of: units, measured/derived, location, instrument (if measured), temporal resolution etc.

- *A parameter table was added to the manuscript (Table 1).*

I found the last paragraph of Methods (p.6 l.1-10) where you define the "seasons": polar night, melt and growing a little disconnected. Obviously, you organise your results according to these categories which I think is nice, but you could add to this description that this is how you will present the data and why this is informative. This would make the 'story' flow a little better.

- *The paragraph was restructured according to the reviewer's comment (p. 6, l. 4-11).*

8. I feel like the conclusions are missing some kind of outlook to what next ie. integrating models to scale results/ investigate other aspects of the enrgy balance or strategies to reduce the energy balance closure problem. I think a few sentences reflecting on ways of building on this study with further work would be useful.

- *The conclusions were updated with outlook and importance of extreme events for modelling purposes and the paper's possible contribution to such modelling tools (p. 15, l. 6-15).*

9. Figure 1a: can you mark the Asiaq station on the map?

- *The figure was updated with the location of the Asiaq-station (see Figure 1a).*

10. Figure 2: mention which site this is in the caption.

- *The figure caption was updated according to the reviewer's comment (see Figure 3).*

11. Picking up on the comments of J . McFadden and while I agree the stacked plots (Figure 3a/b) make it tricky to identify trends in individual years - I think the key point the authors intend to show is the cumulative energy inputs from all components. If that's the intention I would say some form of cumulative presentation is important.

- *The design of the figures was updated to ensure a better readability (see Figure 4).*

12. Not immediately obvious which lines the axis refer to in Figure 6b - soil moisture is indicated on the black line, perhaps can do the same with blue/red lines (evaporation?).

- *The figure was updated to ensure a better readability (see Figure 8).*

This is useful contribution to the arctic surface flux observation literature as the study documents a nice case study of control of the amount of snow (snow depth) on the subsequent evolution of the turbulent surface fluxes and melt. As such, and also given the amount of work it involves to get the data in such an environment, it deserves to be published. I do have some comments however, some major, some minor, that I hope may improve the paper.

Main comments.

First. The paper contains an awful lot of numbers, and no error estimates at all. In the table the SD is given, but if that would be a good measure (and we know it is not), most of the data would fall within the same probability distribution. My first suggestion would be to abandon the use of SD and give the range, and give an error estimate of all your measurements. That helps to assess the significance of your difference.

- *The tables in the manuscript were updated with information on range and error estimates for all the measured parameters (see Table 2-4). Error estimates were considered in the discussion of the revised version of the manuscript (p. 11, chapter 4.1) .*

Second. The energy balance closure is vital to the whole exercise in calculating the amount of energy available for melt as a residual from the energy balance. While I agree that a considerable amount of variability is expected, a value of 67% is very low and needs a little more explaining that referring to site heterogeneity. To show the valid-ity of the eddy covariance measurements I would suggest to include a spectral analysis of the measurements. A good co-spectrum adds to the reliability and acceptance of the data. I also suggest to include this analysis in the description of the methodology, where it belongs, and not include as as an afterthought in the discussion.

- *Sample analysis of cospectra were performed for both sites (p. 4, l. 9-11). The results and discussion of this analysis was added to the manuscript (p. 12, chapter 4.2). Previous cospectra analysis have been performed at the two study sites, showing high-frequency loss of data. However, the range of the high-frequency loss is consistent with values observed from other studies using the same instrumental setup. The information was added and discussed in the manuscript (p. 12, l. 9-20).*

)

Third. Soil moisture. In terms of controlling factors, soil moisture is the key term that controls the partitioning of the energy budget terms. Precipitation and snow depth are just proxies in that sense. I am surprised that only in Figure 6 soil moisture is used. In fact how it is measured is not mentioned at all in $ 2.2. Was it only measured at the dry heath. Please explain and use the data!

- *Soil moisture was only measured at the dry heath since the wet fen is characterized by constant water-saturation. Information on soil moisture sensor and measurement depth is added to the methods section (see chapter 2.2, p. 4, l. 15-16).*

Fourth. Overall the analysis is very descriptive, even lacklustre at times. This is a pity as the data are very valuable! For example when parameters like surface resistance of omega are calculated there is no real effort to explain or interpret these (I find the big difference between the yearly wet fen values somewhat worrying though, given the magnitude of the difference; even between different vegetation types you would not expect such a big difference). This really needs some work. For instance if a wetter Arctic would imply less H, would that provide a negative feedback on the warming trend? There are plenty of such questions to ask given the data and I encourage the authors to think these through and by doing so add more meat to the discussion. I am also surprised to find that there is no mention of the Kasurinen et al., 2014, GCB study at all, given some of the co-authors participated in that study. This does provide a very useful benchmark for the present study.

- *Values of surface resistance were recalculated and checked for their reliability. In the discussion section (chapter 4.2, p. 14, l. 5-17), a paragraph describing possible reasons for the observed differences in surface resistance, omega and Bowen ratio between the two years was added, considering additional literature such as Kasurinen et al., 2014, Lafleur & Rouse, 1988, etc (p 14, l. 5-17)*

Fith. I would rephrase the title to make it a better aligned with the content. Something like a "A comparison of surface energy budgets of : : : in two years with extreme and little snowfall".

- *The title was rephrased. "Two years with extreme and little snowfall: Effects on energy partitioning and surface energy exchange in a high-Arctic tundra ecosystem".*

Minor comments

P1 l10. The use of interannual suggest that many years are used. I would suggest to not use this word and stick to the comparison of a snow rich and snow poor year (see above remark on title as well).

- *The word "interannual" was omitted in the context of the manuscript.*

P3 l27-30. This should be part of the introduction, not site description.

- *The part was moved to the introduction.*

P4 $2.2 Measurements. Have the systems be run side by side in a comparison experiment to show that they provide the same fluxes when at the same site? What are otherwise the errors you expect in the fluxes?

- *No direct comparison experiment of the two systems have been performed. However, we use well-applied sensors and processing schemes which follow the general standards and requirements of ICOS. Measurement errors due to technical sensor specifications at the two sites are therefore considered to be negligible. The information was added to the manuscript (p. 3, l. 29-31, p. 4, l. 9-11).*

P5 l11-13. Is there a way you can quantity the error that this would generate on your estimates of

energy available for melt.

- *We lack information on snow density during that winter. Further, we have no sensors measuring the snowpack temperature when the snow cover is <10 cm. Therefore, we were not able to correct G for storage within the snow pack. However, we assume that heat storage in the snow layer was negligible during that winter, with only little impact on the total energy available for snow melt. The information was added to the manuscript (p. 6, l. 13-15, p. 12, l. 5-8).*

P6 l28-30. I would guess that synoptic variability and weather dynamics also play a role here. You are expressing an extreme 1-D view of the atmosphere here.

- *The paragraph was restructured (p. 7, l. 7-12).*

P7 28-29. This is an example of my first major comment. Are these values really different, or do they fall within you measurement error (given your energy closure for instance)?

- *The related table 3 was updated with information on range and error estimate for the specific parameters.*

P8 l16 gives an another example.

- *The related table 3 was updated with information on range and error estimate for the specific parameters.*

P8 l26-27. Is there any way you can relate this to greenness, density of vegetation as well, or is this really just an effect of the relative contribution of snow versus vegetation. I am asking also in relation to Fig 1, where a different colour seems to be visible for the different years.

- *The paragraph was updated with information on NDVI measurements from the wet fen site (p. 4, l. 20-23, p. 9, l. 12-13).*

p9. $3.3.4. This part really needs some more work and check on the values of Rs in the dry heat growing season. Also it may be better to define a period of maximum gs, rather that show the average which is biased by the shoulder values of the season. Reference here also Kasurinen et al, 2014. This part is presently pretty shallow, I am afraid to say.

- *In the discussion section (see chapter 4.1 and 4.2) we added a paragraph focussing on the interpretation of the presented data (p. 12, l. 4-20).*

P12 l27. I guess you mean soil moisture rather than groundwater? Otherwise how did you determine this?

- *The paragraph was updated to "minerotrophic water supply", p.*

Figure 3a and 3b. Can you add the snowmelt of 3b just to 3a? This avoids repetition of the albedo and snow depth and temperature plot. You adjust the time period to the longer one of 3b.

- *All figures (see Figure 1-9) were updated to provide a better readability.*

Figure 5. Can you reduce the scale of the Y-axis in the lower panel plots so that differences are more visible?

- *The figure was updated according to the reviewer's comment (see Figure 6).*

Figure 6. Can you add a second y-axis that gives the % of use of available energy for the different fluxes?

- *The figure was updated according to the reviewer's comment (see Figure 7).*

Figure 6b is confusing as it does not fit with 6a (it shows both the dry tundra and wet fen and both years) and the structure of all the other plots. Make it a separate plot.

- *The figure was updated according to the reviewer's comment (see Figure 7 and 8).*

This paper reports results from a 2-year field measurement campaign at Zackenberg, Greenland to measure seasonal time series of energy budget components in two con-trasting arctic vegetation types, wet fen and dry heath tundra. The paper makes use of the fact that there was a strong contrast in snow cover between the two years to infer what changes in surface energy exchange might be expected under a warming (and higher precipitation) arctic climate. Changes in the arctic energy budget, especially those studied seasonally, are important. The variety of ecosystem types that have been measured–including in Greenland–is relatively few, making this an important contribu-tion to our knowledge of cold land processes. The manuscript is very clearly written, well organized, and the data are well presented (but see a few comments below). The conclusions are reasonable and are supported by the data. I recommend the paper for publication with only minor revisions. In my view, the paper is very good and the detailed comments below are given as constructive suggestions to improve the final version.

General comments:

1) Figure design. The figures are very nice and clearly labeled; however, some of them are confusing to read. For example, in Figs 3a and 3b the stacked bar plots make it very hard to see the differences between sites and years. Stacked bars are difficult in general, but the large difference in the size of the trends (e.g. the tiny negative values of Rnet) make it really difficult to discern, especially in these small panels. I think the data will come across to the reader better and the paper will have better impact if you can find another way to show these data. Maybe it would be a non-stacked bar plot, a line plot, or something else. Maybe they need to be made larger so that you show only one vegetation type at a time. I understand the reasons why you laid out the panels as you did, it's just that they end up difficult to read and so the result will be better if you can revise them to show the various data values more clearly.

- *All figures were updated (see Figures 1-9).*

Secondly, the use of color is not always allowing the reader to quickly pick out what is what. Sometimes the colors are inconsistent, for example in Fig 5, H and LE are shown in red and blue, and G is in green. But in Fig 6, H and LE follow the same red and blue style, but G is now in gray. Earlier in the paper (Figs 3a and 3b), the same shades of red and blue are used instead for albedo and snow depth. I think you might be better off to remove color from most of the plots where it is possible, and reserve the use of color only where it is the best/only way to show the differences. For example, you could symbolize albedo and snow depth with black lines of different thicknesses, or using dashes. I would suggest placing all of the color figures all together out on a table and then deciding on a system so that the same type of color or symbols are used consistently, and simplify them as much as you can by not using color at all unless it is needed.

- *Figure design and colour regime was updated, using the same or similar colour regime for each parameter in the different figures (see Figures 2-8).*

2) References. The paper does a good job of citing relevant literature overall, but its impact would be improved by making more direct comparisons to other arctic energy budget studies, especially those outside of Greenland. There are still few enough such studies that this is valuable to the scientific community to provide a bigger context for your findings. The manuscript did cite the big review paper of Eugster et al. (2000), but you didn't use that paper to actually compare the energy budget patterns you found to the patterns reported for other arctic ecosystems. At least in the North American Arctic where I have worked, there are other studies that have measured the same energy budget components and even looked at some of the same wet-dry ecosystem comparisons as you are making here. I think the paper would be stronger if it linked into the Arctic literature a little better by directly stating whether the results obtained here are consistent with or different from the values and the overall patterns found in a broader set of arctic sites. This need not be a major addition to the text, but just digging in a bit deeper for each such value or pattern that you highlight in the Discussion.

- *The discussion section (see chapter 4.1 and 4.2) was updated by comparing our measurements with studies from Siberia (Boike et al., 2008), other arctic locations (Sturm and Douglas, 2005; Kasurinen et al., 2014) and previous measurements at the study site (Lund et al., 2014, Soegaard et al., 2001).*

Detailed comments:

Abstract. The last part of the final sentence in the abstract is rather vague. Simply "increasing interannual variability" seems rather general and less interesting that what you actually found. I suggest thinking about how your could "sharpen" this final state-ment to make it more specific and impactful. I could imagine you might want to say something about how interannual climate variability (winter precip) has been shown to have different feed backs to surface energy partitioning depending on ecosystem type...

- *The final section of the abstract was rephrased, focussing on the different effects of ecosystem type and surface properties on surface energy balance.*

p 4, Measurements. Accepted that you are using standard CarboEurope type proce-dures, but can you add a citation to a paper from your flux site that explains site-specific procedures and conditions? If not, then can you please add just a brief description of key site specific information such as how data were screened (spikes, low turbulence, unfavorable meteorological conditions) and how many gaps (what percentage of the record, what percentage of the daytime values that went into the energy budget mea-surements, or whatever you think is relevant)? I see that you later explain that you used the MPI online gap-filling tool to do the gap filling. I am just requesting a little info to characterize data screening and the overall situation with gaps at the two towers. And any other key, site-specific information or procedures that you think are important to the reader. If all of this is summarized in a different flux paper, then it is fine to simply cite it as above, and say that those details are provided in the cited paper.

- *Data processing for both study locations is summarized in Soegaard et al., 2001 and Lund et al. (2012, 2014). The sentence was added to the paragraph (p. 4, l. 6-11).*

p 7, line 10-11: It is very difficult to see the negative values of Rnet in the figure because of its vertical scale and the bar type. Please see general comment above on the figures.

- *The figure was updated (see Figure 4).*

p 8, line 6ff: Please provide a short sentence or two explaining how the 2 study years compare to the long-term climate of the site. Which of the study years was more typical of "mean" climate conditions for this location in terms of the different seasons and meteorological variables? Which aspects were most atypical–only the low-snowfall year, or anything else?

- *The paragraph was updated with information on mean climate conditions for this location and how the two study years compare to the long-term climate of the area (p. 8, l. 17-26).*

p 12, line 4: Recommend changing "shields off" to "reflects".

- *The wording was changed according to the reviewer's comment.*

p 13, Energy Budget Closure section: I suggest that you move the energy budget clo-sure discussion somewhere further up in the Discussion section, rather than as the last item. There are two reasons for this. First, it makes the paper less impactful if the very last item discussed is a technical system-performance assessment like energy budget closure, rather than one of the main findings on your scientific questions. Second, it is arguable that the reader should know whether you think the energy closure imbalance is typical of arctic sites or is in any way a problem before reading onward to the main scientific findings. You might even put it first in the discussion, or at least somewhat closer to the begining of the discussion, and not the very last item.

- *The section on energy budget closure was put first in the discussion (see chapter 4.1).*

In addition, do you think that variations in the depth of permafrost may have been a source of uncertainty/error in estimating the ground heat flux and energy budget closure, and if yes, you could mention that.

- *Information on the impact of permafrost and soil thermal gradients on the ground heat flux was added to the discussion section (p. 12, l. 5-8).*

Finally, I suggest that you end the part about energy budget closure with a comment on how the lack of closure would (or would not) affect your findings. For example, you may believe that the H and LE fluxes are OK and that the closure error is mostly in G. Rn is not affected, and so forth. I think it would make the paper stronger if you could provide a short sentence or 2 at most that interprets what the closure error means for the results you have obtained here.

- *The discussion section was updated with additional information on the accuracy of turbulent heat fluxes and the overall system performance (see chapter 4.1).*

p 14, line 8: Change "discusses" to "discussed"

- *The wording was changed.*

Some other arctic energy budget references (not exhaustive, just some close examples to what you have done here)

Sturm, M., et al. (2005). "Changing snow and shrub conditions affect albedo with global implications." Journal of Geophysical Research-Biogeosciences 110(G01004).

McFadden, J. P., et al. (2003). "A regional study of the controls on water vapor and CO2 exchange in arctic tundra." Ecology 84(10): 2762-2776.

McFadden, J. P., et al. (1998). "Subgrid-scale variability in the surface energy balance of arctic tundra." Journal of Geophysical Research 103(D22): 28947–28961.

Rouse, W. R. (2000). "The energy and water balance of high-latitude wetlands: con-trols and

extrapolation." Global Change Biology 6: 59–68.

Lafleur, P. M. and W. R. Rouse (1988). "The influence of surface cover and climate on energy partitioning and evaporation in a subarctic wetland." Boundary-Layer Meteorol-ogy 44(4): 327–348.

Lafleur, P. M., et al. (1987). "Components of the surface radiation balance of sub-arctic wetland terrain units during the snow-free season." Arctic, Antarctic, and Alpine Research 19(1): 53–63.

- *Additional literature was included into the manuscript.*

---

## Author Response (AR2)

**Technical corrections**

p. 1, l. 23: exerts a strong impact -> exerts a strong influence OR has a strong impact

*The wording was changed to "has a strong impact"*

p. 2, l. 2: and almost doubled compared to the rest of the globe -> and was almost twice as strong as the global average

*The wording was changed according to the editor's comment*

p. 2, l. 20: and THE active layer

*The wording was changed according to the editor's comment*

p. 5, l. 5: "The left side of the equation represents the system's gain of energy and the right-hand side represents the losses of energy." This is not always so? For instance, Rnet is energy loss during the polar night. If you mean this for period average, then please state so. Also please make clear how the signs of the SEB components are defined.

*The paragraph (p. 4, l. 28 – p. 5, l. 3) was updated with information on the signs of the SEB components.*

p. 6, l. 11: equals to -> equals (?)

*The wording was changed to "equals 1.26"*

p. 6, 3.1: Obviously these slopes are significantly smaller than 1. State that this will be discussed in section 4.

*The following sentence was added to the manuscript: "Possible reasons for the observed energy imbalance at both sites are discussed in section 4.1."*

p. 7, l. 14: "The average air temperature, which mainly controls the downwelling longwave radiation..." this is only partly so; emissivity (cloudiness, humidity) also is important. See also next remark.

*The sentence was changed to "The average air temperature was -19.2°C and -15.2°C for the two consecutive polar winter periods and the snow surface temperature was -24.1°C and -19.2°C, respectively."*

p. 7, l. 15/16: " the snow surface temperature together with cloudiness". There appears to be a mashup of information here, please check.

*The sentence was changed to "The average air temperature was -19.2°C and -15.2°C for the two consecutive polar winter periods and the snow surface temperature was -24.1°C and -19.2°C, respectively."*

p.7, l. 21: mainly driven due to -> mainly driven by

*The wording was changed according to the editor's comment.*